# On Smoothness Bounds for Non-Clairvoyant Scheduling with Predictions

**Tianming Zhao**
The University of Sydney
tzha2101@gmail.com

**Albert Y. Zomaya**
The University of Sydney
albert.zomaya@sydney.edu.au

## Abstract

Algorithms with predictions leverage predictions for unknown inputs in online decision-making. These algorithms are analyzed by consistency, i.e., competitive ratio under perfect predictions, and robustness, i.e., competitive ratio under worst-case predictions. Smooth degrading performance with an increased prediction error is also desirable. This paper refines the notion of smoothness, a function of prediction error, defined as the competitive ratio over the problem instances where predictions are guaranteed to provide additional information.

With our refined smoothness metric, we establish smoothness bounds for a few scheduling problems, including online total completion time minimization and makespan minimization. For a single machine to minimize the total completion time, we show a lower bound of $\eta$ and a $\eta^2$-smooth algorithm, where $\eta$ is the prediction error ($\eta \geq 1$); the bound holds for small errors. For parallel identical machines to minimize the makespan, we show a lower bound of $2 - O(\eta^{-2})$ and present an $O(\eta^2)$-smooth algorithm for small errors. Both bounds are tighter than the existing ones. For uniformly-related machines to minimize the makespan, we show a tight lower bound of $\lceil \log \eta \rceil$, matched by an $O(\log \eta)$-smooth algorithm.

## 1 Introduction

Online problems require algorithms to make decisions without complete information. The traditional assumption is that information is either known or unknown, or becomes known after specific events occur (e.g., job size is learned only after the job completion in non-clairvoyant scheduling). Such treatment of information, however, is suboptimal, either too optimistic or too pessimistic. The framework of algorithms with predictions addresses these limitations by assuming that predictions of unknowns are available to support decision-making while admitting potential prediction errors. The framework produces algorithms that leverage additional but possibly inaccurate information against uncertainty. Under this setting, it is desired that the algorithms are *consistent* and *robust*, meaning good performance under perfect and the worst-case predictions. The literature also demands that such algorithms be *smooth* against prediction errors, i.e., the algorithm's performance should degrade gracefully with worsening predictions.

The existing notion of smoothness—competitive ratio written as a function of the prediction error—can sometimes be meaningless. This is because, for a given prediction error, one can often find some problem instances where predictions give no additional information, especially if the error is defined in an "absolute" sense, thus essentially turning the analysis into the cases without predictions. We illustrate this claim with the following example. Consider the problem of single-machine non-clairvoyant scheduling to minimize the total completion time with job size predictions. The job sizes are denoted by $p_1^*, \ldots, p_n^*$ and the predictions $p_1, \ldots, p_n$. Suppose that the prediction error $\eta$ is defined as the $L_1$-distance between the predictions and the true values, i.e., $\sum_{j=1}^n |p_j^* - p_j|$. Now, take any problem instance $I$; one can construct $I'$ by scaling the job sizes such that $\sum_j p_j'^* = \eta$ and setting the predictions all to zero. By construction, the prediction error in $I'$ is $\eta$, but predictions are completely uninformative—any algorithm's performance on $I'$ will match its worst-case performance without predictions. This argument holds for any prediction error $\eta$. Thus, the smoothness curve (i.e., competitive ratio as a function of the prediction error) can be arbitrarily high and thus fail to reflect genuine marginal benefits from reasonable predictions.

We propose a refined definition of smoothness with the intention that any bounds must reveal the marginal benefit of predictions when they do provide additional information. We define smoothness as the competitive ratio over the problem instances where predictions are *guaranteed to provide additional information*. Precise definitions and examples are in the later sections, but the high-level idea is here. We partition the universe of problem instances into (1) ones with perfect predictions, (2) ones with the prediction error $\eta$ large enough so predictions potentially cannot distinguish anything (i.e., the worst-case predictions), and (3) ones with $\eta$ small enough so predictions are guaranteed to add information (i.e., reasonable predictions). Smoothness is then defined as the competitive ratio as a function of the prediction error over the problem instances with reasonable predictions. Consistency is that over the instances with perfect predictions, and robustness is that with the worst-case predictions. To solidify the ideas, we look at familiar non-clairvoyant scheduling with job size predictions. We show that lower bounds on smoothness exist for any deterministic algorithm. Some bounds are asymptotically tight, as algorithms with a matching smoothness guarantee exist. Under our smoothness notion, the lower bounds can be interpreted as the best possible competitive ratio an algorithm can achieve, even if the predictions always provide additional information, which shows the limits and, at the same time, the potential of algorithms with predictions.

We focus on non-clairvoyant scheduling to study smoothness. There have been extensive developments in scheduling theory since its inception (Graham, 1966), offering rich results for clairvoyant and non-clairvoyant scheduling. Algorithms with predictions are a natural advancement to interpolate the existing algorithms, with the hope that the learning component can further improve decision-making and lower bounds of non-clairvoyant scheduling. Scheduling with predictions is also one of the first fields from which algorithms with predictions have grown (Purohit et al., 2018). This study aims to understand the limits and potential of predictions by showing bounds on smoothness when predictions are guaranteed to provide additional information. In particular, we consider (a) single-machine scheduling to minimize the total completion time, (b) parallel identical machines scheduling to minimize the makespan, and (c) uniformly-related machines scheduling to minimize the makespan. These concrete scheduling problems have been studied under algorithms with predictions. Our contributions include providing analysis with a different (perhaps more suitable) prediction error, tighter bounds, algorithms with improved guarantees, and the first established bound.

**Single machine scheduling to minimize the total completion time.** We revisit this problem with a new (to this problem) prediction error metric. The existing works have relied on the $L_1$-distance error, which is scale-sensitive (Purohit et al., 2018; Wei and Zhang, 2020; Bampis et al., 2022). Our error metric, however, is multiplicative without a scaling issue. With this new metric, our analysis for smoothness bound demonstrates a strong min-max game-theoretic dynamic between the scheduler and adversary. The game reveals the edge of the scheduler with predictions: predictions constrain the problem instances the adversary is allowed to use. It turns the adversary to face a constrained optimization, thus forcing a much more involved adversarial strategy to yield any non-trivial bound.

**Parallel machines scheduling to minimize the makespan.** The recent work of Bampis et al. (2023) is the first to consider this problem under algorithms with predictions. The first lower bound has been given, achieved via a static adversary, as well as the first upper bound using *Longest Predicted Job Size First*. We improve the lower bound using a more adaptive adversary that creates problem instances differently according to different prediction errors. In addition, we improve the upper bound by showing an online algorithm based on the offline $(1 + \epsilon)$-approximation algorithm.

**Uniform machines scheduling to minimize the makespan.** The recent work of Zhao et al. (2022) is the first to consider this problem under algorithms with predictions. An $O(\log \eta)$-competitive algorithm has been proposed. However, a lower bound is missing. We fill this gap by giving the first smoothness bound through an adversarial strategy that forces any algorithms to be $\Omega(\log \eta)$-competitive. This bound is asymptotically tight, matching the existing $O(\log \eta)$ competitive ratio.

**Technical contributions.** We introduce a refined definition of smoothness, which measures the performance of an algorithm as predictions worsen when the predictions are guaranteed to provide additional information. For the total completion time minimization with job size predictions, we show a smoothness lower bound of $\eta$ and a $\eta^2$-smooth algorithm for small prediction error $\eta$ ($\leq$ 1.835). For makespan minimization, we show a smoothness lower bound of $2 - O(\eta^{-2})$ for parallel identical machines, and we give a $(\eta^2 + \epsilon)$-smooth algorithm for small $\eta$ ($\leq 1.414$). For uniformly-related machines to minimize the makespan, we show a smoothness lower bound of $\lceil \log \eta \rceil$, which is asymptotically tight, as an $O(\log \eta)$-smooth algorithm exists.

## 2 PRELIMINARIES

We consider non-clairvoyant scheduling with job size predictions. The system consists of $m$ machines and $n$ independent jobs. The number $m$ is not fixed and is part of the problem input. A machine is denoted by $M_i$, $1 \le i \le m$; a job is denoted by $J_j$, $1 \le j \le n$. Job $J_j$ has size $p_j^*$, but this value stays unknown until $J_j$ is completed. We use $P = \max_{1 \le j \le n} p_j^* / \min_{1 \le j \le n} p_j^*$ to denote the maximum job size ratio. Machine $M_i$ has a processing speed $s_i$. For convenience, we sort the machines at a non-increasing speed, i.e., $s_1 \ge s_2 \ge ... \ge s_m$. The machine speeds are known to the scheduler. If job $J_j$ is assigned to machine $M_i$, the processing time for this job will be $p_j^*/s_i$. For the single machine case, there is only one machine with speed one. For parallel identical machines, machines have the same speed, i.e., $s_i = 1, \forall\, i$. For uniformly-related machines, machines are allowed to have different speeds. The jobs are priority-free, meaning there is no dependency between jobs. For completion time minimization, the jobs are preemptive with no cost for preemption. For makespan minimization, jobs are preemptive-restart, meaning that running jobs are non-preemptive but can be canceled and restarted later on any machine with no cost of preemption and migration. The job, however, loses its processing when it is canceled. At any time, a job can be processed on at most one machine, and a machine can process at most one job. All jobs are available at the start. In completion time minimization, our objective is *total completion time*, i.e., $\sum_{j=1}^{n} C_j$. For makespan minimization, our objective is *makespan*, $C_{\max}$, i.e., the time when the last job is completed. In Graham's notation (Graham et al., 1979), our problems are $1 \mid online\text{-}time\text{-}nclv, pmtn \mid \sum C_j$, $Pm \mid online\text{-}time\text{-}nclv, pmtn\text{-}restart \mid C_{\max}$, and $Qm \mid online\text{-}time\text{-}nclv, pmtn\text{-}restart \mid C_{\max}$.

Job size prediction $p_j$ is learned when $J_j$ arrives. Following the works by Lattanzi et al. (2020); Zhao et al. (2024), we define the prediction error for $J_j$ as $\eta_j = \max\left\{p_j/p_j^*, p_j^*/p_j\right\}$, the multiplicative gap between prediction and exact value; the prediction error $\eta$ for a problem instance $I$ is:

$$\eta = \eta(I) = \max_{1 \le j \le n} \eta_j = \max_{1 \le j \le n} \max\left\{p_j/p_j^*, p_j^*/p_j\right\}$$

Observe that $\eta \ge 1$ and predictions are perfect if and only if $\eta = 1$. This multiplicative error has also been used by Azar et al. (2021; 2022). They define the error ($\lambda$) as the product of the maximum underestimation and overestimation factors. The worst-case predictions can have these factors the same, where $\lambda = \eta^2$. We focus on deterministic algorithms and use $A$ to denote an algorithm. A problem instance $I$ consists of the scheduling problem, the number of machines, machine speeds (for uniformly-related machines), job sizes, and job size predictions. Let $\mathcal{I}$ be the collection of all problem instances. Let $A(I)$ and $Opt(I)$ be the cost of the schedule produced by $A$ and an offline optimum. We say an algorithm has a competitive ratio $c$, or is $c$-competitive if $c = \sup_{I \in \mathcal{I}} \frac{A(I)}{Opt(I)}$.

### RELATED WORKS

There has been a growing body of work on algorithms with predictions. For an introduction, we refer to the surveys by Mitzenmacher and Vassilvitskii (2021; 2022). Algorithms with predictions were first introduced in Lykouris and Vassilvtiskii (2018); Purohit et al. (2018) for caching, ski-rental, and scheduling, along with the notions of consistency and robustness. Following these foundational works, many problems have been studied in this framework: caching (Chłedowski et al., 2021), rent-or-buy (Shin et al., 2023), secretary problems (Antoniadis et al., 2020), graphs (Davies et al., 2023), and scheduling (Lattanzi et al., 2020; Azar et al., 2021; Lindermayr et al., 2023). We recommend the online repository (Lindermayr and Megow, 2022a) for a comprehensive overview of the literature.

Completion time scheduling has been studied under algorithms with predictions, including job size predictions (Purohit et al., 2018; Im et al., 2021; Wei and Zhang, 2020; Bampis et al., 2022) and job ordering predictions (Lindermayr and Megow, 2022b; Dinitz et al., 2022). For makespan minimization with predictions, Lattanzi et al. (2020) proposed a competitive algorithm with machine load predictions for unrelated machines; it was later improved by Li and Xian (2021) to attain asymptotic optimality. For uniformly-related machines scheduling with job size predictions, Zhao et al. (2022) gave a competitive algorithm for static scheduling, and later, the algorithm was extended to dynamic scheduling (Zhao et al., 2024). For parallel identical machines, Bampis et al. (2023) gave the first lower and upper bounds. Additionally, there have been efforts to address the limitations of assessing algorithms solely via consistency and robustness. Azar et al. (2023) proposed the metric of discrete-smoothness, which evaluates performance as a function of prediction error but has the restriction that predictions must have the same form as the solution to an optimization problem.

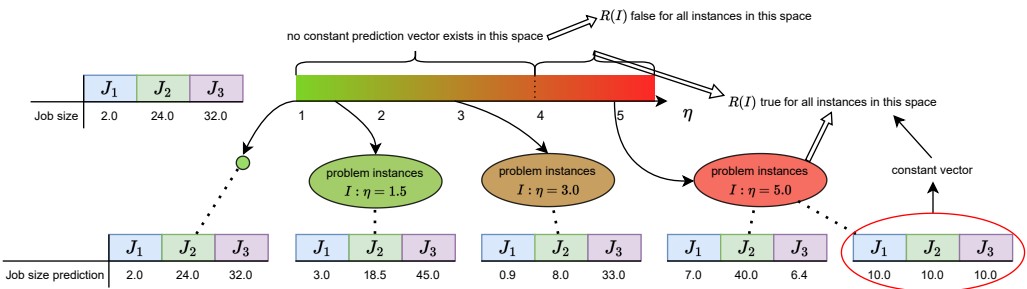

Figure 1: A visualization of predicate $R(I)$. Consider a set of three jobs: $J_1$, $J_2$, and $J_3$ with job sizes 2, 24, and 32. Let $\eta(I) = \max_{1 \leq j \leq n} \max \left\{ p_j/p_j^*, p_j^*/p_j \right\}$. There are problem instances (vectors of job size predictions) generated per level of prediction error $\eta$. We show a few examples for $\eta = 1, 1.5, 3.0, 5.0$. For $\eta < 4$, there exists no constant prediction vector, meaning $R(I)$ is false for all instances with $\eta < 4$, i.e., the prediction error in this range is reasonable. For $\eta \geq 4$, there will always exist a constant prediction vector for every error level. Thus, $R(I)$ is true for any instance with $\eta \geq 4$, i.e., the prediction error is too large. The interpretation is that, if $\eta \geq 4$ for this job set, the predictions may provide no information by showing a constant vector. For the analysis of smoothness, we will focus on cases where predictions do add information.

## 3    CONSISTENCY, ROBUSTNESS, AND SMOOTHNESS

This section introduces consistency, robustness, and our refined version of smoothness. The idea is to partition the universe of problem instances $\mathcal{I}$ into three sets $\mathcal{I}_c, \mathcal{I}_r, \mathcal{I}_s \subseteq \mathcal{I}$. Consistency, robustness, and smoothness are then defined to be the worst-case performance ratio over $\mathcal{I}_c$, $\mathcal{I}_r$, and $\mathcal{I}_s$, respectively. Set $\mathcal{I}_c$ contains problem instances with no prediction error; $\mathcal{I}_r$ contains problem instances where the error is large enough so that the predictions provide no additional information in the worst case; $\mathcal{I}_s$ contains instances where the error guarantees additional information.

We use bold symbol to indicate a ($n$-dimensional) vector, so $\boldsymbol{p^*} = (p_j^*)_{1 \leq j \leq n}$ denotes a vector of job sizes and $\boldsymbol{p} = (p_j)_{1 \leq j \leq n}$ job size predictions. Define error function $\eta(\boldsymbol{p}, \boldsymbol{p^*})$ to be the prediction error of $\boldsymbol{p}$ with respect to $\boldsymbol{p^*}$, e.g., $\eta(\boldsymbol{p}, \boldsymbol{p^*}) = \max_{1 \leq j \leq n} \max \left\{ p_j/p_j^*, p_j^*/p_j \right\}$. For problem instance $I$, we use $p^*(I)$ and $p(I)$ to denote the vectors of job sizes and job size predictions in $I$, and $\eta(I) = \eta(p(I), p^*(I))$. We assume $\eta$ is non-decreasing with worsening predictions, i.e., a larger error means worse.

We will introduce a predicate $R(I)$ to model the worst-case predictions. It takes a problem instance as input and outputs true or false, indicating whether the predictions reveal no additional information. Before presenting its definition, we give its high-level intuition. For a fixed prediction error $\eta$, if one can create the job size predictions with 1) error equal to $\eta$, and 2) all predictions have the same value, the predictions will not allow one to partially order jobs, or even partition them according to the predictions. That is, the predictions are as if they are not present. In other words, for the given predictions, the set of possible actual job size vectors includes a constant one, meaning that all permutations of the jobs according to their job sizes remain possible. In such a scenario, the predictions reveal no additional information. Define predicate $R(I)$ as: "$\exists$ constant $\boldsymbol{p} : \eta(\boldsymbol{p}, p^*(I)) = \eta(I)$." The predicate checks if any constant vector of job size predictions exists with error $\eta(I)$, where a constant vector means any arbitrary element is the same value. Intuitively, the predictions reveal no additional information in the worst case if $R(I)$ holds. We partition problem instances using $R(I)$.

**Definition 3.1** (Problem instances partitions)**.**

$$\mathcal{I}_c = \{I | \eta(I) = \eta(p^*(I), p^*(I))\}, \ \mathcal{I}_r = \{I | R(I), I \notin \mathcal{I}_c\}, \ \mathcal{I}_s = \{I | \neg R(I), I \notin \mathcal{I}_c\}$$

**Example 3.1.** *If* $\eta(I) = \max_{1 \leq j \leq n} \max \left\{ p_j/p_j^*, p_j^*/p_j \right\}$*, we have:* $\mathcal{I}_c = \{I | \eta(I) = 1\}, \mathcal{I}_r = \{I | \eta(I) \geq \sqrt{P(I)}, \eta(I) > 1\}, \mathcal{I}_s = \{I | 1 < \eta(I) < \sqrt{P(I)}\}$*, where* $P(I) = \max_{p_j^*, p_k^* \in p^*(I)} p_j^*/p_k^*$*. Figure 1 visualizes the notion of predicate* $R(I)$ *under this error metric.*

The definitions of $\mathcal{I}_c$, $\mathcal{I}_r$, and $\mathcal{I}_s$ may change under different error metrics and can also be extended to other predictions not just for job sizes. The partition is complete, i.e., $\mathcal{I}_c \cup \mathcal{I}_r \cup \mathcal{I}_s = \mathcal{I}$, and disjoint, i.e., $\mathcal{I}_c \cap \mathcal{I}_r = \mathcal{I}_c \cap \mathcal{I}_s = \mathcal{I}_r \cap \mathcal{I}_s = \emptyset$, giving clear-cuts between the problem instances considered by these metrics. We have developed a way to partition problem instances according to the prediction error. We then define consistency, robustness, and smoothness based on this partition.

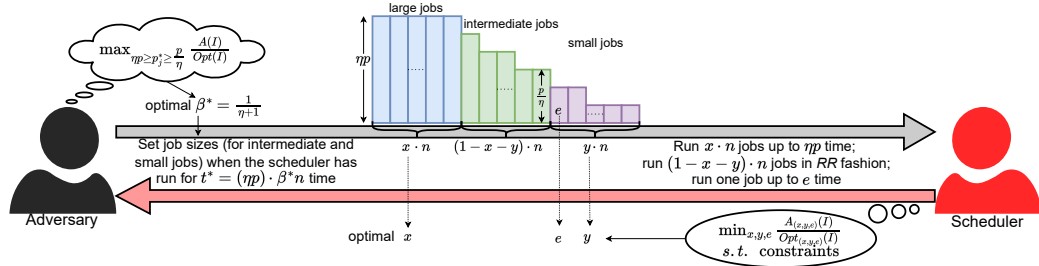

Figure 2: A visualization of the analysis for smoothness bounds for single machine completion time scheduling. The adversary decides the job size distribution using optimization (1). The heights of the bars (jobs) represent the job sizes. Given such an adversary, the best any scheduler can do is to run optimization (2) and schedule jobs accordingly. We show that the lower bounds derived from such dynamics are as stated in Theorem 4.1.

**Definition 3.2** (Consistency, robustness, and smoothness). For a minimization problem, we use $c_A$, $r_A$, and $s_A(\eta)$ to denote consistency, robustness, and smoothness of algorithm $A$, defined as:

$$\sup_{I \in \mathcal{I}_c} \frac{A(I)}{Opt(I)}, \ \sup_{I \in \mathcal{I}_r} \frac{A(I)}{Opt(I)}, \ \sup_{I \in \mathcal{I}_s, \eta(I) \leq \eta} \frac{A(I)}{Opt(I)}$$

where $A(I)$ and $Opt(I)$ denote the costs of the schedules produced by algorithm $A$ and the optimum.

Note that smoothness $s_A(\eta)$ is a function of $\eta$ over the problem instances where the predictions must always provide additional information. One can view the metrics consistency, robustness, and smoothness as the competitive ratios of an algorithm over subsets of problem instances partitioned by the prediction error. Here, we discuss the connection between our proposed smoothness and the notion of the competitive ratio degrading smoothly with the prediction error, which exists before this work. The existing notion requires an error-dependent competitive ratio, i.e., a bound written as a function of the error, to be derived for an algorithm. Our smoothness explicitly defines such a function. One can then say that an algorithm is smooth if its smoothness is in a low-order form of $\eta$, e.g., $\log \eta$ or a polynomial in $\eta$. In addition, the lower bounds indicate the limit of how such smoothness can go. If a $\Omega(\log \eta)$ bound exists, for example, it is impossible to attain a better asymptotic functional form for the competitive ratio as a function of the error, even if the predictions are revealing additional information. Such explicit smoothness allows one to study the limit and potential of the informational advantages carried by the predictions for learning-augmented algorithms.

The authors believe smoothness is a better metric (than consistency and robustness) to consider. This is because, in practice, one will likely be dealing with predictions having some errors but not adversarial, so neither consistency nor robustness best describes the performance that matches the practice. While it is hard to know the prediction error apriori, it is not so to estimate the range of errors (e.g., through historical backtesting), so practitioners can optimize smoothness over the estimated range of errors. Another advantage is on lower bounds. Roughly speaking, lower bounds indicate the informational advantages carried by the predictions; they inform the marginal value of improving the accuracy of predictive models from a business's perspective. To see this, suppose we establish a smoothness lower bound of $\Omega(\eta)$ for some problem; that means improving the prediction error by a factor of a half may potentially give a doubled improvement on the end goal of the business; this may inform the business decision of allocating more resources to building accurate predictive models. We proceed with showing smoothness bounds for some scheduling problems.

## 4 Single machine to minimize the total completion time

This section focuses on single-machine scheduling to minimize the total completion time. Some notable prior results on completion time scheduling with predictions include a consistent and robust algorithm for the single machine case (Purohit et al., 2018), optimal robustness-consistency trade-offs (Wei and Zhang, 2020), an extension to multiple machines (Bampis et al., 2022), and the analysis for the case with partial predictions (Benomar and Perchet, 2024). All these studies rely on the prediction error measured by the $L_1$-distance between the predictions and the actual values, i.e., $\sum_{j=1}^{n} |p_j^* - p_j|$. One drawback of the $L_1$-distance metric is scale-sensitive. Consider two instances: $I_1$ with $p_1^* = 1, p_2^* = \ldots = p_n^* = 2$ and $p_1 = 1, p_2 = \ldots = p_n = 3$ and $I_2$ with

$p_1^* = 1, p_2^* = \ldots = p_n^* = 2 \cdot 100$ and $p_1 = 1, p_2 = \ldots = p_n = 3 \cdot 100$. Roughly, $I_2$ is 100-time scaled $I_1$. The $L_1$-distance will report a 100 times larger prediction error, despite that the competitive ratio remains if the scheduled job order remains the same. The multiplicative-based metric will report an error of 1.5 for both cases, indifferent to the algorithm from the prediction error's perspective. Thus, it is interesting to understand the bounds of algorithms with predictions under the multiplicative measure for errors. An upper bound that can be immediately obtained from Azar et al. (2022) is $O(\eta^2 \log \eta)$. We show the following lower and (improved) upper bounds on smoothness.

**Theorem 4.1.** *If $A$ is an algorithm for a single machine to minimize total completion time and $\eta^*$ is the positive root of $x^4 - 4x - 4 = 0$ ($\eta^* \approx 1.835$), $s_A(\eta) \geq \eta$ for $\eta \leq \eta^*$, $s_A(\eta) \geq \lambda(\eta)$ for $\eta^* < \eta \leq 1 + \sqrt{3}$, and $s_A(\eta) \geq 2$ for $\eta > 1 + \sqrt{3}$, where $\lambda(\eta) = \frac{2\eta\left(\eta^2 + \eta + \sqrt{(\eta+1)(2\eta^4 + \eta^3 + 9\eta^2 + 16\eta + 8)}\right)}{\eta^4 + 4\eta^2 + 8\eta + 4}$.*

**Theorem 4.2.** *For minimizing the total completion time, there exists a $\eta^2$-competitive algorithm.*

*Remark* 4.3. We have $\lambda(\eta) \geq 0.18\eta + 1.5$ for $\eta^* < \eta \leq 1 + \sqrt{3}$, so the piecewise smoothness bound yields $s_A(\eta) \geq 0.18\eta + 1.5$ over $(\eta^*, 1 + \sqrt{3}]$, a simplified lower bound for $\lambda(\eta)$ in Theorem 4.1.

*Remark* 4.4. A visualization of the smoothness bounds in Theorems 4.1 and 4.2 is in Figure 3.

The proofs of Theorems 4.1 and 4.2 are given in the appendix. We provide an overview of the analysis here. Fix any $\eta$. It is natural to guess that *Shortest Predicted Job Size First (SPJF)*, i.e., running jobs in non-decreasing order of job size predictions, is optimal for small prediction errors and that *Round-Robin (RR)*, i.e., equally sharing the processor among the active jobs, is optimal for large errors. We will show an adversary strategy to attack any algorithm under the constraint that the predictions guarantee to provide additional information. To do so, we reveal the true size of the smallest job while ensuring that the rest are indistinguishable via predictions. Consider problem instance $I$: the first $n$ jobs have non-increasing sizes with the same job size prediction $p$, and the last job has a tiny job size but the correct job size prediction. We obtain: (1) predictions reveal information, (2) the first $n$ jobs are indistinguishable, and (3) the smallest job, although visible, contributes insignificantly to the total completion time. The benefit, however, that comes with predictions is that the indistinguishable jobs cannot differ too much from each other by size; they are constrained by the error $\eta$. Since the smallest job cannot impact the total completion time much, we give any algorithm an advantage by finishing this job for free, and may ignore it to simplify the discussion.

Any algorithm $A$ will operate blindly against the first $n$ jobs, so the adversary can force the algorithm to schedule these jobs in non-increasing order of job sizes. The optimum is running the jobs in non-decreasing order of job sizes. Then, the adversary considers the following optimization:

$$\max_{\eta \cdot p \geq p_1^* \geq p_2^* \geq \ldots \geq p_n^* \geq \frac{p}{\eta}} \quad \frac{A(I)}{Opt(I)} = \frac{\sum_{j=1}^n (n-j+1) \cdot p_j^*}{\sum_{j=1}^n j \cdot p_j^*} \tag{1}$$

which is a linear-fractional program with linear constraints, where the maximum occurs at a vertex of the feasible polyhedron. The relevant theory for this type of optimization problem can be found in Boyd and Vandenberghe (2004). The optimum is to set every job size to either one end of the extremes ($\eta \cdot p$ or $p/\eta$). Therefore, the optimization reduces to finding a $\beta, 0 \leq \beta \leq 1$, to maximize the ratio, where $p_j^* = \eta \cdot p, j \leq \beta n$ and $p_j^* = p/\eta, j > \beta n$. Let $f(\beta)$ be the ratio after substituting this optimal solution structure. Taking the limit as $n \to \infty$ and maximizing the function over $\beta \in [0, 1]$, we find the maximum achieved at $\beta^* = \frac{1}{\eta+1}$. This gives us some hint on the adversarial strategy. However, a straightforward adversary that assigns $\beta^* n$ jobs of size $\eta p$ (large jobs) and the remaining $(1 - \beta^*)n$ jobs of size $p/\eta$ (small jobs) is not enough to yield the bound in Theorem 4.1. To see this, consider a two-phase algorithm $B$. In phase one, $B$ processes $p/\eta$ time for each job. During this phase, any small jobs processed will be completed, leaving large jobs. Phase two runs the rest of the jobs sequentially. A direct calculation shows $\lim_{n \to \infty} \frac{B(I)}{Opt(I)} < \eta$, for $\eta > 1.6$.

A more adaptive adversary works as follows. Let any algorithm $A$ runs for $t^* = (\eta p) \cdot \beta^* n$ time. A job is completed if and only if it has been processed for $\eta p$ time. These jobs are called *large* jobs. At time $t^*$, the adversary lets any job that has been processed for $p/\eta$ to finish immediately (called *intermediate* jobs) and sets the rest of the jobs to have job size $p/\eta$ (*small* jobs). We show that when the prediction error is small, this adversary pushes the algorithm to complete all the large jobs before $t^*$ and to leave no intermediate jobs. This implies the lower bound of $\eta$. When the error is large, the adversary pushes the algorithm to equally share the processor among the active jobs. This implies the lower bound of 2. When the error is moderate, however, the algorithm may mix *SPJF* and *RR* to fight against this adversary. In this case, the lower bound implied by this adversary is $\lambda(\eta)$.

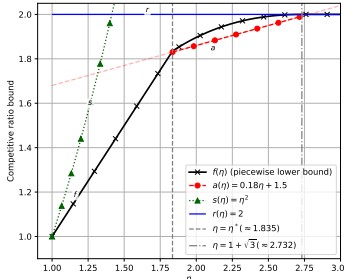
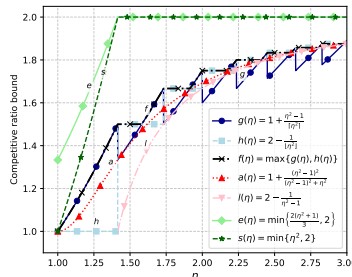

Figure 3: Functions related to smoothness bounds in Section 4. Function $f$ is the piecewise lower bound in Theorem 4.1; $a$ is its linear first-order approximate over $(\eta^*, 1+\sqrt{3}]$ in Remark 4.3; $s$ is the upper bound in Theorem 4.2.

Figure 4: Function $f$ is the lower bound in Theorem 5.1; $g, h$ are its components; $a, l$ are the bounds in Remark 5.2; $s$ is the upper bound in Theorem 5.4; $h$ and $e$ are the bounds in Bampis et al. (2023); they should compare with $f$ and $s$.

We outline the approach to derive the bound implied by the above adversary. Suppose algorithm $A$ knows the adversary's strategy and the value of $\eta$ upfront. Effectively, $A$ has to decide: (1) the number of large jobs to finish before time $t^*$, denoted by $x \cdot n$, $0 \leq x \leq 1$, (2) the number of small jobs to leave at $t^*$, denoted by $y \cdot n$, $0 \leq y \leq 1$, and (3) the total time allocated to the unfinished small job group before $t^*$, denoted by $e$. We can show that these three parameters $(x, y, e)$ represent all candidates for the optimal online schedule against this adversary. This claim follows from two observations about the optimal schedule. The first is that the number of intermediate jobs is determined by $(1-x-y) \cdot n$, with a total of $t^* - (\eta p) \cdot xn - e$ time allocated to them, and it is optimal for $A$ to equally share the processor among all the intermediate jobs. The second observation is that it is optimal for $A$ to bound $0 \leq e < p/\eta$ because the time allocated to small jobs can be aggregated and used to yield more intermediate jobs, which favors $A$ if $e \geq p/\eta$. Then, it is optimal for $A$ to concentrate $e$ unit of time on one small job before $t^*$. Let $I$ be the adversarial instance. We can calculate $A_{(x,y,e)}(I)$ and $Opt_{(x,y,e)}(I)$. Then, algorithm $A$ considers the following optimization:

$$\min_{x,y,e} \frac{A_{(x,y,e)}(I)}{Opt_{(x,y,e)}(I)} \tag{2}$$

$$\text{s.t. } 0 \leq x \leq \frac{1}{1+\eta}, \max\left\{0, \eta^2 x + (1-x) - \frac{\eta^2}{\eta+1} + \frac{e\eta}{pn}\right\} \leq y \leq \frac{\eta}{\eta+1}, 0 \leq e < \frac{p}{\eta}$$

where the constraint for $y$ comes from that every intermediate job must have a size at least $p/\eta$. Let $\frac{2\eta(\eta)}{p} \cdot h(x,y,t)$ be the coefficient of the highest order term in $A_{(x,y,e)}(I) - t \cdot Opt_{(x,y,e)}(I)$. Showing some $M$ is a lower bound for $\lim_{n\to\infty} \frac{A_{(x,y,e)}(I)}{Opt_{(x,y,e)}(I)}$ reduces to $h(x, y, M) \geq 0$. Observe $h$ is strictly convex in $x$ and $y$. We can show that $h \geq 0$ indeed holds for the bounds stated in Theorem 4.1. Figure 2 gives a visualization of the above discussion.

We show that *SPJF* achieves a competitive ratio of $\eta^2$, which proves Theorem 4.2. The proof follows the standard technique for analyzing *SPJF* used by Purohit et al. (2018). Unfortunately, showing a better bound is non-trivial, although our lower bound analysis indicates that *SPJF* attains a competitive ratio $\eta$ against the proposed adversary. Closing this gap is of future research interest.

## 5 PARALLEL IDENTICAL MACHINES TO MINIMIZE THE MAKESPAN

This section focuses on parallel identical machines scheduling to minimize the makespan. The clairvoyant case has been solved by a polynomial approximation scheme (Hochbaum and Shmoys, 1987), where a $(1 + \epsilon)$-competitive algorithm is given with run time of $O((n/\epsilon)^{1/\epsilon^2})$. The run time has then been improved to $O((n/\epsilon)^{(1/\epsilon)\log(1/\epsilon)})$ by Leung (1989). There is a lower bound of one on consistency, which, as we will show, can be matched up to any precision. The non-clairvoyant case has been solved by Graham's list scheduling (Graham, 1969), which is a $(2 - \frac{1}{m})$-competitive algorithm. List scheduling is optimal, given a lower bound of $(2 - \frac{1}{m})$ on the competitive ratio (Shmoys et al., 1995). Thus, a lower bound of $(2 - \frac{1}{m})$ on robustness exists, matched by list scheduling. So far, we have not considered the predictions that are between perfect and worst-case.

The first lower and upper bounds on algorithms with predictions are given by Bampis et al. (2023). They show a lower bound of $2-1/\lfloor\eta^2\rfloor$ and an upper bound of $\min\{2(\eta^2+1)/3, 2\}$, i.e., $2(\eta^2+1)/3$ if $\eta^2 < 2$ or 2 otherwise. The lower bound follows from a static adversary instance; the upper bound is obtained by *Longest Predicted Job Size First* based on *Longest Job Size First (LJSF)* (Graham, 1969), which is $4/3$-consistent and 2-robust. We will improve both bounds. In Theorem 5.1, we show a dynamic adversary based on the prediction error, which yields a better lower bound of $\max\{2 - 1/\lfloor\eta^2\rfloor, 1 + (\eta^2 - 1)/\lceil\eta^2\rceil\}$. In Theorem 5.4, we combine the $(1 + \epsilon)$-approximation algorithm, i.e., the approximate offline optimum, and list scheduling to achieve a competitive ratio of roughly $\min\{\eta^2, 2\}$. It implies a better upper bound as $\eta^2 < 2(\eta^2 + 1)/3$ if $\eta^2 < 2$. Moreover, our algorithm achieves the optimal 1-consistency up to any precision while retaining the optimal 2-robustness. Figure 4 shows a visual comparison of these bounds.

**Theorem 5.1.** *If A is an algorithm to minimize the makespan,* $s_A(\eta) \geq \max\left\{1 + \frac{\lfloor\eta^2\rfloor - 1}{\lfloor\eta^2\rfloor}, 1 + \frac{\eta^2 - 1}{\lceil\eta^2\rceil}\right\}$.

We give intuitions for the proof of Theorem 5.1. We want to construct problem instances where predictions reveal minimal but additional information. Observe that the makespan is heavily determined by the allocation of large jobs: the competitive ratio improves from $2 - \frac{1}{m}$ to $\frac{4}{3} - \frac{1}{3}m$ with *LJSF*. Thus, our strategy, similar to the previous section, is to reveal the true size of the smallest job while ensuring that the rest are indistinguishable via predictions. This ensures (1) $\eta(I) < \sqrt{P(I)}$ so predictions reveal information, and (2) we can analyze the marginal gain of the predictions. We consider the following setup: $J_1$ has the largest job size, $\eta^2$, with a prediction of $\eta$; $J_2, ..., J_{n-1}$ have the same job sizes close to 1 with predictions of $\eta$; $J_n$ has a very small job size, $\epsilon$, with a prediction of $\epsilon$. In this case, jobs $J_1, ..., J_{n-1}$ are indistinguishable. Although the smallest job is visible, it contributes insignificantly to the makespan. Again, the algorithm benefits from that the indistinguishable jobs cannot differ too much by size; they are constrained by the error $\eta$. Given that the smallest job will not impact the makespan much, we ignore it to simplify the discussion.

Consider the simplest adversary strategy with 3 jobs and 2 machines. Job $J_1$ has a size $\eta^2$ with prediction $\eta$, and jobs $J_2$ and $J_3$ have size 1 with prediction $\eta$. These jobs are indistinguishable, so we can force any algorithm to allocate jobs $J_2$ and $J_1$ to machine $M_1$ and $J_3$ to $M_2$. The resulting makespan is $1 + \eta^2$, while the optimum allocates $J_1$ to $M_1$ and $J_2$, $J_3$ to $M_2$, attaining a makespan of 2. We have established a lower bound of $\frac{1+\eta^2}{2}$ if $\eta^2 < 2$. When $\eta^2 \geq 2$, we set the sizes of $J_2$ and $J_3$ to $\frac{\eta^2}{2}$, so the construction bounds the smoothness by $\frac{3}{2}$. We can have a better adversary strategy as $\eta$ increases above $\frac{5}{2}$. Consider 7 jobs and 3 machines: $p_1^* = \eta^2$ and $p_j^* = 1$ for $2 \leq j \leq 7$, with $p_j = \eta$ for all $j$. We can force any algorithm to allocate $J_1, J_2, J_3$ to $M_1$, $J_4, J_5$ to $M_2$, and $J_6, J_7$ to $M_3$, achieving a makespan of $\eta^2 + 2$, while the optimum is 3. This yields a lower bound $\frac{\eta^2+2}{3}$, higher than $\frac{3}{2}$ if $\eta^2 > \frac{5}{2}$. Our proof of Theorem 5.1 generalizes this adversary strategy for any $\eta$.

Observe that the lower bound on smoothness does not exceed 2. This is consistent with the 2-competitiveness of list scheduling, i.e., a 2-competitive algorithm exists for any prediction error. The following bound is useful to represent the asymptotic behavior of smoothness.

*Remark* 5.2. $s_A(\eta) \geq \max\left\{1 + \frac{\lfloor\eta^2\rfloor - 1}{\lfloor\eta^2\rfloor}, 1 + \frac{\eta^2 - 1}{\lceil\eta^2\rceil}\right\} \geq 1 + \frac{(\eta^2 - 1)^2}{(\eta^2 - 1)^2 + \eta^2} \geq 2 - \frac{1}{\eta^2 - 1} \in 2 - O(\eta^{-2})$

We then give a consistent, smooth, and robust algorithm. Our algorithm has a competitive ratio of $\min\{\eta^2 + \epsilon, 2 - \frac{1}{m}\}$ with a run time polynomial in $n$ for any given $\epsilon > 0$. The algorithm has near optimal consistency of $1 + \epsilon$, smoothness of $\eta^2 + \epsilon$ for $\eta^2 < 2$, and the optimal robustness of $2 - \frac{1}{m}$. Our analysis builds on the following result for clairvoyant scheduling.

**Theorem 5.3** (Hochbaum and Shmoys (1987); Leung (1989)). *There exists a $(1+\epsilon)$-approximation algorithm with run time $O((n/\epsilon)^{(1/\epsilon \log(1/\epsilon))})$ for clairvoyant scheduling for any given $\epsilon > 0$; it determines the allocation of jobs to machines upfront.*

A straightforward application of this $(1+\epsilon)$-approximation is to run the approximation algorithm as if the job size predictions are actual job sizes. Call this algorithm *SIMPLE*. *SIMPLE* has consistency $1 + \epsilon$ and smoothness $O(\eta^2)$, as each job is under-/over-estimated by at most a factor of $\eta$. The algorithm, however, is not robust—it has the robustness of $\Theta(m)$, the worst competitive ratio attainable by allocating all the jobs to any machine. To see why *SIMPLE* is not robust, create $n = Km + m - 1$ jobs for some large enough $K$; $m - 1$ of them are small jobs with job size prediction 1 and job size $\delta$ for some small $\delta > 0$ and the rest $Km$ are large jobs and have job size prediction $2\epsilon$ with job size

1 for some small positive $\epsilon$ ($\epsilon < \frac{1}{2Km}$). For any given $\epsilon$, *SIMPLE* must allocate each small job on a separate machine and all the large jobs on one machine—it is a $(1 + \epsilon)$-approximation with respect to job size predictions. The makespan achieved by *SIMPLE* is $Km$ while the optimum allocates the large jobs evenly and has a makespan at most $K + \delta$. Thus, the robustness of *SIMPLE* is $\Theta(m)$.

To overcome the poor robustness of *SIMPLE*, we propose a simple technique called *list-adjusting*: if some machine $M_i$ finishes its last job determined by *SIMPLE* when some other waiting jobs are allocated to other machines, we run any of them on $M_i$ instead of idling it. This technique ensures that the final schedule is replicable by list scheduling (thus the name list-adjusting), where list scheduling is $(2 - \frac{1}{m})$-competitive for any job order. Note that list-adjusting needs *SIMPLE* to have a static schedule to follow upfront, so knowing if a job is the last on some machine is possible.

**Theorem 5.4.** *For parallel identical machines to minimize the makespan, there exists a* $\min\{\eta^2 + \epsilon, 2 - \frac{1}{m}\}$*-competitive algorithm with run time polynomial in* $n$ *for any given* $\epsilon > 0$.

Figure 4 plots the functions related to the smoothness bounds we have derived, including the lower bound $f$ (Theorem 5.1) and upper bound $s$ (Theorem 5.4), and compares them with the bounds derived by Bampis et al. (2023). Functions $g$ and $h$ take alternating domination in $f$. Function $a$ is a (non-trivial) unified lower bound for $f$, which can be well-approximated by function $l$ for large $\eta$. The optimal smoothness should lie between functions $s$ and $f$. Therefore, a gap exists between the upper and lower bounds for smoothness. Closing this gap for such a fundamental scheduling problem may advance our knowledge about the smoothness of algorithms with predictions.

# 6 UNIFORMLY-RELATED MACHINES TO MINIMIZE THE MAKESPAN

This section focuses on uniformly-related machines. This problem has recently been studied under the algorithms with predictions setting; an $O(\min\{\log \eta, \log m\})$-competitive algorithm has been given by Zhao et al. (2024). The literature, however, is missing a lower bound on smoothness. With our definition of smoothness, we show that a $\lceil \log \eta \rceil$ smoothness lower bound exists; this bound is asymptotically tight, given that an $O(\log \eta)$-smooth algorithm exists. Our key result is the following.

**Theorem 6.1.** *If* $A$ *is an algorithm to minimize the makespan,* $s_A(\eta) \geq \lceil \log \eta \rceil$.

Our adversary strategy has two key elements. The first is revealing additional but minimum information via predictions. We allow predictions to reveal the size of the smallest job while leaving the rest indistinguishable. The second is pushing the larger jobs to be finished later. Both elements follow the idea used in the previous cases. We show that this strategy implies a $\Omega(\log \eta)$ lower bound.

We create our adversarial problem instance for any given prediction error $\eta$. Part of the design is indistinguishable jobs, for which we follow a family of instances developed by Cho and Sahni (1980) that has been shown effective in attacking a non-clairvoyant scheduler (Shmoys et al., 1995). Let $k = \lceil 2 \log \eta \rceil + 1$. Create $k$ groups of machines $G_i$ and $k$ groups of jobs $T_i$, for $0 \leq i \leq k$. Every machine in $G_i$ has a speed of $2^i$, and every job in $T_i$ has size $2^i$, for $0 \leq i < k$, i.e., the machines in the same group are identical to each other, and the same for jobs. Every job in groups $T_0, ..., T_{k-1}$ has job size prediction of $\eta$, and $|G_i| = |T_i| = 2^{2k-2i-1}$, for $0 \leq i < k$. The $k$-th group is special, with one machine in $G_k$ and one job in $T_k$. The speed of the machine in $G_k$ is $\delta$, and the size of the job in $T_k$ is $\delta$ with job size prediction of $\delta$, for a small $\delta < \min\{\frac{1}{\lceil \log \eta \rceil}, \frac{1}{2}\}$. In this instance, we have $n = m$. Index the jobs and machines such that $p_1^* \leq p_2^* \leq ... \leq p_{n-1}^*$, $p_n^* = \delta$, $s_1 \leq s_2 \leq ... \leq s_{m-1}$, $s_m = \delta$, $\bigcup_{i=0}^{k-1} G_i = \{M_1, M_2, ..., M_{m-1}\}$, and $\bigcup_{i=0}^{k-1} T_i = \{J_1, J_2, ..., J_{n-1}\}$. The job size predictions are: $p_j = \eta$, for $1 \leq j \leq n-1$ and $p_n = \delta$. Table 1 summarizes the problem instance.

There are a few properties of this problem instance: (1) the prediction error is $\eta$; (2) the instance belongs to $\mathcal{I}_s$; (3) the optimal makespan is $C_{\max}^* = 1$; (4) any schedule that completes a job other than $J_n$ on $M_m$ must have a makespan at least $\frac{1}{\delta} > \lceil \log \eta \rceil$. Thus, Theorem 6.1 is trivial if the scheduler completes a job other than $J_n$ on $M_m$. We assume $J_1, ..., J_{n-1}$ are completed on $M_1, ..., M_{m-1}$ for the following discussion so the scheduler correctly allocates $J_n$.

| mach grps.: | $G_0$ | $G_1$ | ... | $G_{k-1}$ | $G_k$ |
|---|---|---|---|---|---|
| job grps.: | $T_0$ | $T_1$ | ... | $T_{k-1}$ | $T_k$ |
| $s_i / p_j^*$: | $2^0$ | $2^1$ | ... | $2^{k-1}$ | $\delta$ |
| $\lvert G_i \rvert / \lvert T_i \rvert$: | $2^{2k-1}$ | $2^{2k-3}$ | ... | $2^1$ | $1$ |
| $p_j$: | $\eta$ | $\eta$ | ... | $\eta$ | $\delta$ |

Table 1: The adversarial problem instance.

We introduce the notion of *adversarial properties* and show that the adversary can force the scheduler to finish larger jobs later. For any schedule, let $t_s(j)$ and $t_f(j)$ be the last starting time and the finishing time of job $J_j$. A job can be preempted and later restarted; $t_s(j)$ denotes the last starting time. Let $m(j)$ be the machine that completes $J_j$. Let $j_i^{\max}$ be the highest index for the jobs in $T_i$.

**Lemma 6.2.** *Any scheduler can be forced to construct a schedule with the adversarial properties:*

1. $t_f(1) \le t_f(2) \le \cdots \le t_f(n-1)$
2. $t_s(l) + \frac{p_j^*}{s_{m(l)}} \ge t_f(j), \quad \forall\, j < l < n$

We show that, against the adversary problem instance, any schedule with the adversarial properties has a makespan at least $\lceil \log \eta \rceil$ through the following bounds. Lemma 6.3 shows that finishing any group of jobs costs at least one unit of time. Lemma 6.4 bounds the time difference between finishing two groups of jobs. Putting these bounds together yields Theorem 6.1.

**Lemma 6.3.** $\max_{J_j \in T_i} \{ t_f(j_i^{\max}) - t_s(j) \} \ge 1, \forall\, 0 \le i < k$, *under the adversarial properties.*

**Lemma 6.4.** $t_f(j_i^{\max}) - t_f(j_{i-1}^{\max}) \ge \frac{1}{2}, \forall\, 1 \le i < k$, *under the adversarial properties.*

The bound in Theorem 6.1 is asymptotically tight, as an $O(\min\{\log \eta, \log m\})$-competitive algorithm exists (Zhao et al., 2024). With a $\Omega(\log m)$ non-clairvoyant lower bound (Shmoys et al., 1995), the existing algorithm achieves asymptotically optimal consistency, smoothness, and robustness. This shows the possibility of attaining optimality in all three metrics.

## 7 DISCUSSION

Our definition of smoothness excludes the problem instances with predictions that reveal no additional information. An alternative way to define smoothness would be to include these instances, i.e, $s'_A(\eta) = \sup_{I \in \{I | I \notin \mathcal{I}_c\}, \eta(I) \le \eta} \frac{A(I)}{Opt(I)}$. While this also separates smoothness and robustness, we choose our definition because 1) it fully separates the problem instances considered by the two metrics, 2) the smoothness lower bound has the interpretation that "no algorithm can achieve better than $s_A(\eta)$ even if the predictions are revealing additional information", and 3) the smoothness bounds are stronger. All the derived bounds (Theorems 4.1, 4.2, 5.1, 5.4, and 6.1) still hold for $s'_A(\eta)$, because the problem instances considered in our definition are a subset of those in $s'_A(\eta)$. It remains open if the proposed smoothness definition makes a difference in the upper bounds, i.e., if there exists an algorithm $A$ to some problem where $s_A(\eta) < s'_A(\eta)$.

We highlight that our proposed notion of smoothness applies to other online problems. Different online problems and prediction settings require different interpretations of predicate $R(I)$ in the definition. Take the one-way trading problem as an example (Elenter et al., 2024; Benomar and Perchet, 2025). Following the notations in Elenter et al. (2024), one candidate definition of $R(I)$ is $|\hat{p} - p_\sigma^*| \ge \max(p_\sigma^* - 1, M - p_\sigma^*)$, where $\sigma$ is the input sequence of exchange rates, $p_\sigma^*$ the maximum rate in $\sigma$, $\hat{p}$ the maximum-rate prediction, $|\hat{p} - p_\sigma^*|$ the prediction error, and $M$ an upper bound on the rates known upfront. The intuition is that if $R(I)$ holds for some problem instance $I$, $p_\sigma^*$ can take any value in $[1, M]$, given $\hat{p}$, which effectively gives no additional information, as we know $p_\sigma^* \in [1, M]$ upfront. This predicate $R(I)$ allows one to partition the problem instances, thus leading to concrete definitions of consistency, smoothness, and robustness following the setup in our work.

## 8 CONCLUSIONS

Smoothness bounds are derived for some non-clairvoyant scheduling with predictions under our refined smoothness metric, which forces additional information from predictions. In the competitive analysis, we see that predictions constrain the adversary, forcing it to have a much more involved adversarial strategy. Lower bounds exist in smoothness—the informational advantage of predictions cannot be unbounded, but gaps exist between the lower and upper bounds for virtually all problems considered. It will be interesting to see if any algorithm can achieve (exactly) optimal consistency, robustness, and smoothness. A positive answer will uncover the full potential of algorithms with predictions. A negative answer may reveal the ultimate trade-offs in this model.

ACKNOWLEDGMENTS

We thank the anonymous reviewers for their feedback and suggestions, which have improved this work. Professor Albert Zomaya and Dr. Tianming Zhao would like to acknowledge the support of the Australian Research Council Research Hub for Future Digital Manufacturing (IH230100013). Tianming expresses his gratitude to his family. He is grateful to his wife, Min Li, and his son, Michael Renqian Zhao, whose love and support have been a constant source of motivation and inspiration. He also warmly acknowledges the forthcoming new addition to his family.

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

## A  PROOFS FOR SECTION 3

*Proof for Example 3.1.* It is easy to verify for $\mathcal{I}_c$, as $\eta(p^*(I), p^*(I)) = 1$. Observe that $\mathcal{I}_r \cap \mathcal{I}_s = \mathcal{I}_r \cap \mathcal{I}_c = \mathcal{I}_c \cap \mathcal{I}_s = \emptyset$ and $\mathcal{I}_r \cup \mathcal{I}_s = \mathcal{I} - \mathcal{I}_c$. It suffices to show that $\sqrt{P(I)}$ is the boundary for $\eta(I)$ to separate $\mathcal{I}_r$ and $\mathcal{I}_s$ under $\eta(I) > 1$. Consider any problem instance $I$ with $\eta = \eta(I) \geq \sqrt{P(I)}$ and $\eta(I) > 1$. Create a constant vector of predictions $\boldsymbol{p^c} = (p^*_{\min} \cdot \eta)_{1 \leq j \leq n}$, where $p^*_{\min} := \min_{1 \leq j \leq n} p^*(I)$. Observe that $\eta(\boldsymbol{p^c}, p^*(I)) = \eta$. This means $I \in \mathcal{I}_r$.

Consider any problem instance $I$ with $\eta(I) < \sqrt{P(I)}$ and $\eta(I) > 1$. Without loss of generality, let $p^*_1$ and $p^*_2$ be the minimum and maximum job sizes. For any constant predictions vector $\boldsymbol{p} = (p)_{1 \leq j \leq n}$ with some constant $p$, it holds that $\frac{p^*_2}{p} \cdot \frac{p}{p^*_1} = \frac{p^*_2}{p^*_1} = P(I)$, which means one of $\frac{p^*_2}{p}$ or $\frac{p^*_2}{p^*_1}$ is at least $\sqrt{P(I)}$. Therefore, $\eta(\boldsymbol{p}, p^*(I)) \geq \sqrt{P(I)} > \eta(I)$. This holds for all constant vectors. Thus, predicate $R(I)$ is not true. This means $I \in \mathcal{I}_s$. $\qquad\square$

## B  PROOFS FOR SECTION 4

This section proves the following theorems.

**Theorem B.1** (Theorem 4.1)**.** *If $A$ is an algorithm for a single machine to minimize total completion time, $s_A(\eta) \geq \eta$ for $\eta \leq \eta^*$, $s_A(\eta) \geq \lambda(\eta)$ for $\eta^* < \eta \leq 1 + \sqrt{3}$, and $s_A(\eta) \geq 2$ for $\eta > 1 + \sqrt{3}$, where $\eta^*$ is the positive root of $x^4 - 4x - 4 = 0$ ($\eta^* \approx 1.835$), and $\lambda(\eta) = \frac{2\eta\left(\eta^2 + \eta + \sqrt{(\eta+1)(2\eta^4 + \eta^3 + 9\eta^2 + 16\eta + 8)}\right)}{\eta^4 + 4\eta^2 + 8\eta + 4}$.*

**Theorem B.2** (Theorem 4.2)**.** *For a single machine to minimize the total completion time, there exists a $\eta^2$-competitive algorithm.*

Table 2 presents the notations used in the analysis.

| Symbol | Meaning / Definition |
|---|---|
| $n$ | Number of jobs. |
| $J_j$ | Job $j$, $1 \leq j \leq n$. |
| $p^*_j$ | True job size of job $J_j$. |
| $p_j$ | Job size prediction for $J_j$. |
| $P$ | Maximum job size ratio $P = (\max_j p^*_j)/(\min_j p^*_j)$. |
| $C_j$ | Completion time of job $J_j$. |
| $\eta_j$ | Prediction error for job $J_j$: $\eta_j = \max\{p_j/p^*_j, p^*_j/p_j\}$. |
| $\eta$ | Total prediction error $\eta = \max_j \eta_j$. |
| $A$ | Deterministic online algorithm under analysis. |
| $Opt$ | Offline optimum schedule. |
| $I$ | A problem instance. |
| $\eta(I)$ | Total prediction error of $I$. |
| $P(I)$ | Maximum job size ratio of $I$. |
| $A(I), Opt(I)$ | Objective value of $A$ and $Opt$ on instance $I$. |
| $s_A(\eta)$ | Smoothness of $A$ as a function of $\eta$. |
| $\eta^*$ | Positive root of $x^4 - 4x - 4 = 0$; numerically $\eta^* \approx 1.835$. |
| $\lambda(\eta)$ | Intermediate lower-bound function defined in Theorem 4.1. |
| $\beta^*$ | Fraction of "large" jobs in the adversary construction; $\beta^* = 1/(\eta+1)$. |
| $t^*$ | Cut-off time $(\eta p)\beta^* n$ used by the adversary. |
| $x, y, e$ | Schedule parameters: finished-large fraction, leftover-small fraction, and extra time spent on one small job before $t^*$. |
| $h(x, y, t)$ | The expression governing the leading term of $A(I) - t \cdot Opt(I)$ in the analysis. |

Table 2: Notations used in the analysis.

PROOF OUTLINE

We outline the proof first. Fix any $\eta$. We reveal the actual size of the smallest job while ensuring that the rest are indistinguishable via predictions. Consider the following problem instance $I$: jobs have non-increasing sizes $p_1^* \geq p_2^* \geq \ldots \geq p_n^*$, where $p_j^* \in [p/\eta, \eta p], p_j = p, \forall 1 \leq j \leq n$, for some large positive $p$, and a tiny job $J_{n+1}$ with $p_{n+1}^* = p_{n+1} = \epsilon \ll \min\{\frac{p}{\eta^3}, \frac{p}{\eta n+1}\}$. Then, we have: (1) $\eta(I) < \sqrt{P(I)}$ so predictions reveal information, (2) jobs $J_1, \ldots, J_n$ are indistinguishable via predictions, and (3) the smallest job, although visible, contributes insignificantly to the total completion time. Given that the smallest job does not impact the total completion time and the competitive ratio much, we give any algorithm an advantage by finishing this job for free and may ignore it to simplify the discussion.

Any algorithm $A$ will operate blindly against the indistinguishable jobs $J_1, \ldots, J_n$, so the adversary can force the algorithm to schedule the jobs in non-increasing order of job sizes. The optimum is running the jobs in non-decreasing order of job sizes. Thus, we have $A(I) = \sum_{j=1}^n (n - j + 1) \cdot p_j^*$ and $Opt(A) = \sum_{j=1}^n j \cdot p_j^*$. Then, the adversary considers the following optimization:

$$\max_{\eta \cdot p \geq p_1^* \geq p_2^* \geq \ldots \geq p_n^* \geq \frac{p}{\eta}} \quad \frac{A(I)}{Opt(I)} = \frac{\sum_{j=1}^n (n - j + 1) \cdot p_j^*}{\sum_{j=1}^n j \cdot p_j^*}$$

which is a linear-fractional program with linear constraints, where the maximum occurs at a vertex (boundary) of the feasible polyhedron. Therefore, the optimization reduces to finding a $\beta, 0 \leq \beta \leq 1$, to maximize the ratio, where $p_j^* = \eta \cdot p, j \leq \beta n$ and $p_j^* = p/\eta, j > \beta n$. Let $f(\beta)$ be the ratio after substituting this optimal solution structure. Taking the limit as $n \to \infty$, we have:

$$\lim_{n \to \infty} f(\beta) = \frac{\eta^2(2\beta - \beta^2) + (1 - \beta)^2}{\eta^2 \beta^2 + (1 - \beta^2)}$$

Maximizing the right-hand side over $\beta \in [0, 1]$, we find the maximum achieved at $\beta^* = \frac{1}{\eta+1}$. However, a straightforward adversary that assigns $\beta^* n$ jobs of size $\eta p$ (large jobs) and the remaining $(1 - \beta^*)n$ jobs of size $p/\eta$ (small jobs) is not enough to yield the bound. To see this, consider a two-phase algorithm $B$. In phase one, $B$ processes $p/\eta$ time for each job sequentially. During this phase, any small jobs processed will be completed, leaving large jobs. Phase two runs the rest of the jobs sequentially. A direct calculation shows $B(I) = \frac{np}{2\eta(\eta+1)^2}((2\eta^2 + 4\eta + 1)n + \eta^3 + 2\eta^2 - 1)$ and $Opt(I) = \frac{np(2n+\eta+1)}{2(\eta+1)}$. We have $\lim_{n \to \infty} \frac{B(I)}{Opt(I)} = 1 + \frac{1}{\eta} - \frac{1}{2\eta^2 + 2\eta}$, strictly less than $\eta$ for $\eta > 1.6$.

The adversary works as follows. Let any algorithm $A$ runs for $t^* = (\eta p) \cdot \beta^* n$ time. A job is completed (immediately) if and only if it has been processed for $\eta p$ time. These jobs are called *large* jobs. At time $t^*$, the adversary sets $p_j^* = x_j(t^*) + \varepsilon$ if job $J_j$ has been processed for $x_j(t^*) \geq p/\eta$ time with a very small positive $\varepsilon$ (called *intermediate* jobs), or $p_j^* = p/\eta$ if $x_j(t^*) < p/\eta$ (*small* jobs). We show that when the prediction error is small, i.e., $\eta \leq \eta^* (\approx 1.835)$, this adversary pushes the algorithm to complete all the large jobs before $t^*$ and to have no intermediate jobs. This implies the lower bound of $\eta$. On the other hand, the strategy pushes the algorithm to equally share the processor among the active jobs when the error is large, i.e., $\eta > 1 + \sqrt{3} (\approx 2.732)$. This implies the lower bound of 2, matched by *RR*. When the error is moderate (i.e., $\eta^* < \eta \leq 1 + \sqrt{3}$), however, the algorithm may mix *SPJF* and *RR* to fight against this adversary if it knows $\eta$. In this case, the lower bound implied by this strategy is $\lambda(\eta)$, somewhere between $\eta^*$ and 2.

We outline the approach to derive the bound implied by the above adversary. Suppose algorithm $A$ knows the adversary's strategy and the value of $\eta$ upfront. Effectively, $A$ has to decide: (1) $x \cdot n$, $0 \leq x \leq 1$, the number of large jobs to finish before time $t^*$, (2) $y \cdot n, 0 \leq y \leq 1$, the number of small jobs to leave at $t^*$, and (3) $e$, the total time allocated to the unfinished small job group before $t^*$. We can show that these three parameters $(x, y, e)$ represent all candidates for the optimal online schedule against this adversary. This claim follows from two observations about the optimal schedule. The first is that the number of intermediate jobs is determined by $(1 - x - y) \cdot n$, with a total of $t^* - (\eta p) \cdot xn - e$ time allocated to them, and it is optimal for $A$ to equally share the processor among all the intermediate jobs, i.e., every intermediate job has size $\frac{t^* - (\eta p) \cdot xn - e}{(1 - x - y) \cdot n}$. The second observation is that it is optimal for $A$ to bound $0 \leq e < p/\eta$ because the time allocated to small

jobs can be aggregated and used to yield more intermediate jobs, which favors $A$ if $e \geq p/\eta$. Then, it is optimal for $A$ to concentrate $e$ unit of time on one small job before $t^*$. Let $I$ be the adversarial instance. We have: $A_{(x,y,e)}(I) = (\frac{1}{2}\eta p x^2 + \beta^* \eta p(1-x) + \frac{1}{2\eta}py^2) \cdot n^2 + (\frac{1}{2}\eta px + \frac{1}{2\eta}py - ey) \cdot n$ and $Opt_{(x,y,e)}(I) = \left( \frac{1}{2\eta}py(2-y) + \frac{1}{2}\eta p[x^2 + (\beta^* - x)(1+x-y)] \right) \cdot n^2 + (\frac{1}{2}\eta p \beta^* + \frac{1}{2\eta}py - \frac{1}{2}e(1+x-y)) \cdot n - \frac{1}{2}e$. Then, algorithm $A$ considers the following optimization:

$$\min_{x,y,e} \frac{A_{(x,y,e)}(I)}{Opt_{(x,y,e)}(I)}$$

$$\text{s.t. } 0 \leq x \leq \frac{1}{1+\eta}, \max\left\{0, \eta^2 x + (1-x) - \frac{\eta^2}{\eta+1} + \frac{e\eta}{pn}\right\} \leq y \leq \frac{\eta}{\eta+1}, 0 \leq e < \frac{p}{\eta}$$

where the bound for $y$ comes from that every intermediate job must have a size of at least $p/\eta$. Let $\frac{2\eta(\eta)}{p} \cdot h(x,y,t)$ be the coefficient of $n^2$ (i.e., the highest order term) in $A_{(x,y,e)}(I) - t \cdot Opt_{(x,y,e)}(I)$; we have: $h(x,y,t) = y^2 (\eta t + \eta + t + 1) + y \left(-\eta^3 tx - \eta^2 tx + \eta^2 t - 2\eta t - 2t\right) + (\eta^3 tx + \eta^3 x^2 - \eta^2 t + \eta^2 x^2 - 2\eta^2 x + 2\eta^2)$. Showing some bound $M$ is the minimum for $\lim_{n\to\infty} \frac{A_{(x,y,e)}(I)}{Opt_{(x,y,e)}(I)}$ reduces to showing $h(x,y,M) \geq 0$. Observe that $h$ is strictly convex in $x$ and $y$. Therefore, with any fixed $x$, the minimum of $h$ occurs at either the boundaries of $y$ or at $\tilde{y} = \frac{t(\eta^3 x + \eta^2 x - \eta^2 + 2\eta + 2)}{2(\eta+1)(t+1)}$. Similarly, with any fixed $y$, the minimum of $h$ occurs at either the boundaries of $x$ or $\tilde{x} = \frac{\eta ty - \eta t + ty + 2}{2(\eta+1)}$. Given these candidates for the minimum, we can show that $h(x,y,\eta) \geq 0$ for $\eta \leq \eta^*$, $h(x,y,\lambda(\eta)) \geq 0$ for $\eta^* < \eta \leq 1 + \sqrt{3}$, and $h(x,y,2) \geq 2$ for $\eta > 1 + \sqrt{3}$, which completes the proof of the Theorem.

PROOF OF THEOREM 4.1

We introduce the adversarial strategy first.

**Adversarial strategy:** Fix any $\eta > 1$. Pick a large enough $p$ and $n$. The adversary presents the following problem instance $I$: jobs have non-increasing sizes $p_1^* \geq p_2^* \geq \ldots \geq p_n^*$, where $p_j^* \in [p/\eta, \eta p], p_j = p, \forall 1 \leq j \leq n$, and a tiny job $J_{n+1}$ with $p_{n+1}^* = p_{n+1} = \epsilon \ll \min\{\frac{p}{\eta^3}, \frac{p}{\eta n+1}\}$. It lets any algorithm $A$ processes jobs for $t^* = (\eta p) \cdot \beta^* n$ time, where $\beta^* = 1/(\eta + 1)$. A job is completed (immediately) if and only if it has been processed for $\eta p$ time. These jobs are called *large* jobs. At time $t^*$, the adversary sets $p_j^* = x_j(t^*) + \varepsilon$ if job $J_j$ has been processed for $x_j(t^*) \geq p/\eta$ time with a very small positive $\varepsilon \to 0$ (called *intermediate* jobs), or $p_j^* = p/\eta$ if $x_j(t^*) < p/\eta$ (*small* jobs). The very small positive $\epsilon$ and $\varepsilon$ are set so that $(\epsilon + \varepsilon)n$ is less than $\frac{p}{\eta}$ times any small positive constant; then, the contributions of $\epsilon$ and $\varepsilon$ to the competitive ratio can be controlled to any arbitrary precision. Therefore, we treat them as $0$ in the analysis without affecting the results. The number of jobs $n$ will be set large and properly enough so that $\beta^* n$ is (at least approximately) an integer. This adversarial strategy satisfies the requirements for smoothness analysis.

**Lemma B.3.** $1 < \eta(I) < \sqrt{P(I)}$.

*Proof.* Observe that $\max_j p_j^* \geq p/\eta$ while $\min_j p_j^* = \epsilon < p/\eta^3$. Therefore, $P(I) \geq \frac{p/\eta}{\epsilon} > \frac{p/\eta}{p/\eta^3} = \eta^2 \Rightarrow \eta < \sqrt{P(I)}$. The lemma holds with $\eta > 1$ by the choice of $\eta$. $\square$

Consider any scheduler $A$. We give $A$ two advantages: (1) after time $t^*$, $A$ schedules the jobs optimally, i.e., it knows the remaining job sizes at time $t^*$ so it can run the jobs via shortest remaining job size first, and (2) after $A$ decides the parameters $(x, y, e)$, it sequentially runs the $xn$ large jobs first in the $t^*$ period. Therefore, the following analysis operates on a lower bound of $A(I)$.

**Scheduler $A$:** Suppose algorithm $A$ knows the adversary's strategy and the value of $\eta$ upfront. Effectively, $A$ has to decide: (1) $x \cdot n, 0 \leq x \leq 1$, the number of large jobs to finish before time $t^*$, (2) $y \cdot n, 0 \leq y \leq 1$, the number of small jobs to leave at $t^*$, and (3) $e$, the total time allocated to the unfinished small job group before $t^*$. Since we treat $\epsilon = \varepsilon = 0$, the scheduler finishes the tiny job for free and all the intermediate jobs at $t^*$. From $t^*$ onward, the scheduler runs the remaining small jobs via the shortest remaining job size first. Lemma B.4 shows that these three parameters $(x, y, e)$

represent all candidates for the optimal online schedule against this adversary. We will later refine the ranges for these three parameters.

**Lemma B.4** (Canonical representation). *Fix $\eta > 1$ and let I be the adversarial instance described above. For any online deterministic algorithm A, there exists an algorithm with the same or smaller competitive ratio, where any schedule produced by which is determined by three parameters:*

$$x, y, e \quad \text{satisfying} \quad 0 \leq x \leq 1, \ 0 \leq y \leq 1, \ 0 \leq e < p/\eta,$$

*where (1) $xn$ large jobs are already completed at time $t^*$, (2) $yn$ small jobs have received $e$ units of processing in total; that $e$ is concentrated on one of them, and (3) the remaining $(1 - x - y)n$ intermediate jobs have each received exactly $\frac{t^* - (\eta p) \cdot xn - e}{(1 - x - y)n}$ units of processing by $t^*$.*

*Thus, the triple $(x, y, e)$ enumerates all candidates for an optimal schedule against the adversary.*

This claim follows from two observations (Lemmas B.5 and B.6) about the optimal schedule. The first is that the number of intermediate jobs is determined by $M = (1 - x - y)n$, with a total of $T_M = t^* - (\eta p) \cdot xn - e$ time allocated to them, and it is optimal for $A$ to equally share the processor among all the intermediate jobs, i.e., every intermediate job has size $\frac{T_M}{M}$. The second observation is that it is optimal for $A$ to bound $0 \leq e < p/\eta$ because the time allocated to small jobs can be aggregated and used to yield more intermediate jobs, which favors $A$ if $e \geq p/\eta$. Then, it is optimal for $A$ to concentrate $e$ unit of time on one small job before $t^*$.

**Lemma B.5** (Equal–share property for intermediate jobs). *Let A be any deterministic online algorithm and $\sigma$ be the schedule it generates up to time $t^*$ against the adversary. Denote by $I(\sigma)$ the problem instance the adversary finalizes after observing $\sigma$. Assume $\sigma$ is described by $(x, y, e)$ of Lemma B.4. Then there exists another algorithm $A'$ that coincides with A except that, during $[0, t^*]$, it shares the processor uniformly among the M intermediate jobs, giving each exactly $T_M/M$ units of processing. Let $\sigma'$ be the resulting schedule (up to time $t^*$) and $I(\sigma')$ the corresponding instance. We have:*

$$A\big(I(\sigma)\big) = A'\big(I(\sigma')\big), \qquad Opt\big(I(\sigma)\big) \leq Opt\big(I(\sigma')\big),$$

*and therefore:*

$$\frac{A'\big(I(\sigma')\big)}{Opt\big(I(\sigma')\big)} \leq \frac{A\big(I(\sigma)\big)}{Opt\big(I(\sigma)\big)}.$$

*Hence, without loss of generality, we may restrict attention to algorithms whose schedules equal–share the processor among intermediate jobs by time $t^*$.*

*Proof.* Let $w_1, \ldots, w_M$ be the amounts of processing given to the $M$ intermediate jobs by time $t^*$ in $\sigma$, so $\sum_{i=1}^{M} w_i = T_M$. Without loss of generality, we sort $w_1 \leq \cdots \leq w_M$. Let $s_1, \ldots, s_S$, $(0 \leq s_j < p/\eta)$ be the processing each of the $S = yn$ small jobs has received; thus $\sum_{j=1}^{S} s_j = e$. After observing $\sigma$ at time $t^*$, the adversary finalizes the true job sizes and the remaining work is:

$$\underbrace{(p/\eta - s_1), \ldots, (p/\eta - s_S)}_{S \text{ items}}.$$

**1. The algorithm's cost does not depend on the individual $w_i$.**

Observe that the contribution of the intermediate jobs is $t^* M$ to the total completion time, which does not depend on the vector $(w_1, \ldots, w_M)$. After the intermediate jobs are gone, the instance consists solely of the $S$ small jobs whose remaining sizes are $(p/\eta - s_1), \ldots, (p/\eta - s_S)$; these quantities are independent of the $w_i$. Therefore the cost of $A$ that comes from the small jobs depends on $(s_1, \ldots, s_S)$ but again not on $(w_1, \ldots, w_M)$. Combining the parts,

$$A\big(I(\sigma)\big) = C_{\text{large}} + t^* M + F\big(s_1, \ldots, s_S\big),$$

where $C_{\text{large}}$ is the cost of large jobs completed by $t^*$ and $F$ is a function of the small–job residual vector only. Hence $A\big(I(\sigma)\big)$ depends on $M$ and the multiset $\{s_j\}$, but not on the vector $(w_1, \ldots, w_M)$. Therefore, for any other schedule $\sigma'$ with the same $(x, y, e)$ (hence the same $M$ and $\{s_j\}$), we have:

$$A\big(I(\sigma)\big) = A'\big(I(\sigma')\big)$$

**2. Equalizing $w_i$ increases the offline optimum.**

The offline optimum runs the jobs in non-decreasing order. Under the adversary's sizes that order is:

$$\underbrace{p/\eta, \ldots, p/\eta,}_{S \text{ small jobs}} \underbrace{w_1, \ldots, w_M}_{M \text{ intermediate jobs}}, \underbrace{\eta p, \ldots, \eta p,}_{L \text{ large jobs}}$$

where $L = xn$. Let $\phi(\boldsymbol{w}) = \sum_{j=1}^{M}(M - j + 1)\, w_j$. Write:

$$Opt(I(\sigma)) = C_{\text{small}} + C_{\text{large}} + \tfrac{p}{\eta}SM + \phi(\boldsymbol{w})$$

where $C_{\text{small}} = \frac{p}{\eta}\frac{S(S+1)}{2}$, $C_{\text{large}} = (\frac{p}{\eta}S + T_M)L + \eta p\frac{L(L+1)}{2}$, and $\frac{p}{\eta}SM$, all depending on $M, S, L, T_M$ but independent of how the total $T_M$ is distributed among the $w_i$. We bound $\phi(\boldsymbol{w})$ by Chebyshev sum inequality for oppositely ordered sequences:

$$\phi(\boldsymbol{w}) = \sum_{j=1}^{M}(M - j + 1)\, w_j \leq \sum_{j=1}^{M}(M - j + 1) \cdot \left(\frac{\sum_{j=1}^{M} w_j}{M}\right) = \frac{M+1}{2}T_M$$

The maximum is achieved if $w_j = \sum_{j=1}^{M} w_j/M = T_M/M, \forall\, j$, achieved by $A'$. Therefore,

$$Opt\big(I(\sigma')\big) = C_{\text{small}} + C_{\text{large}} + \tfrac{p}{\eta}SM + \frac{M+1}{2}T_M \geq Opt\big(I(\sigma)\big)$$

Putting parts 1 and 2 together proves the claim. $\qquad\square$

**Lemma B.6** (Total processor allocation to small jobs). *Let $A$ be any deterministic online algorithm that equal-share processing for intermediate jobs and $\sigma$ be the schedule it generates up to time $t^*$ against the adversary. Denote by $I(\sigma)$ the problem instance finalized after observing $\sigma$. Assume $\sigma$ is described by the triple $(x, y, e)$ of Lemma B.4 with $e \geq p/\eta$. Write $e = z \cdot p/\eta + e'$, where $z$ is integer and $0 \leq e' < p/\eta$. Then there exists another algorithm $A'$ that coincides with $A$ except that, during $[0, t^*]$, it allocates $e'$ time on one of the small jobs and $0$ time on the other small jobs and sets $yn - z$ small jobs, i.e., it increases $z$ intermediate jobs. Effectively, the characteristic triple for the schedule generated by $A'$ is $(x, y - z/n, e')$. Let $\sigma'$ be the resulting schedule (up to time $t^*$) and $I(\sigma')$ the corresponding instance. We have:*

$$A\big(I(\sigma)\big) \geq A'\big(I(\sigma')\big), \qquad Opt\big(I(\sigma)\big) \leq Opt\big(I(\sigma')\big),$$

*and therefore:*

$$\frac{A'\big(I(\sigma')\big)}{Opt\big(I(\sigma')\big)} \leq \frac{A\big(I(\sigma)\big)}{Opt\big(I(\sigma)\big)}.$$

*Hence, without loss of generality, we may restrict attention to algorithms whose schedules allocate $e < p/\eta$ time to small jobs and to just one of them by time $t^*$.*

*Proof.* Let $w_1, \ldots, w_M$ be the amounts of processing given to the $M = (1 - x - y)n$ intermediate jobs by time $t^*$ in $\sigma$, so $\sum_{i=1}^{M} w_i = T_M$. Let $s_1, \ldots, s_S, (0 \leq s_j < p/\eta)$ be the processing each of the $S = yn$ small jobs has received; thus $\sum_{j=1}^{S} s_j = e$. Without loss of generality, we sort $w_1 \leq \cdots \leq w_M$ and $s_1 \leq \cdots \leq s_S$. After observing $\sigma$ at time $t^*$, the adversary finalizes the true job sizes and the remaining work is:

$$\underbrace{(p/\eta - s_1), \ldots, (p/\eta - s_S)}_{S \text{ items}}.$$

Let $L = xn$ and $T_M = t^* - (\eta p) \cdot xn - e$. We can write:

$$A\big(I(\sigma)\big) = \underbrace{\eta p\,\frac{L(L+1)}{2}}_{\text{large}} + \underbrace{t^* M}_{\text{intermediate}} + St^* + \underbrace{\sum_{j=1}^{S} j\left(\tfrac{p}{\eta} - s_j\right)}_{\text{small}}$$

and

$$Opt(I(\sigma)) = \underbrace{\frac{p}{\eta}\frac{S(S+1)}{2}}_{\text{small}} + \underbrace{\frac{p}{\eta}SM + \frac{T_M}{M}\frac{M(M+1)}{2}}_{\text{intermediate}} + \underbrace{(\frac{p}{\eta}S + T_M)L + (\eta p)\frac{L(L+1)}{2}}_{\text{large}}$$

For $A'$, the number of large jobs remain $L = xn$, the number of small jobs becomes $S' = S - z = y'n = yn - z$, and $M' = M + z = (1 - x - y)n + z$, and $T'_M = T_M + z\frac{p}{\eta}$. After observing $\sigma'$ at time $t^*$, the adversary finalizes the true job sizes and the remaining work is:

$$\underbrace{p/\eta, \ldots, p/\eta, (p/\eta - e')}_{S' \text{ items}}.$$

We write:

$$A'\big(I(\sigma')\big) = \underbrace{\eta p\,\frac{L(L+1)}{2}}_{\text{large}} + \underbrace{t^*M'}_{\text{intermediate}} + \underbrace{S't^* + \frac{p}{\eta}\frac{S'(S'+1)}{2} - e'S'}_{\text{small}}$$

and

$$Opt(I(\sigma')) = \underbrace{\frac{p}{\eta}\frac{S'(S'+1)}{2}}_{\text{small}} + \underbrace{\frac{p}{\eta}S'M' + \frac{T'_M}{M'}\frac{M'(M'+1)}{2}}_{\text{intermediate}} + \underbrace{(\frac{p}{\eta}S' + T'_M)L + (\eta p)\frac{L(L+1)}{2}}_{\text{large}}$$

We have:

$$A\big(I(\sigma)\big) - A'\big(I(\sigma')\big)$$
$$= (M - M')t^* + (S - S')t^* + \frac{p}{\eta}\frac{S(S+1)}{2} - \sum_{j=1}^{S} j s_j - \frac{p}{\eta}\frac{S'(S'+1)}{2} + e'S'$$
$$= \frac{p}{\eta}\frac{(2S - z + 1)z}{2} + S'e' - \sum_{j=1}^{S} j s_j$$
$$= \underbrace{\left[\frac{p}{\eta}(S + \cdots + (S - z + 1)) + (S - z)(e - z\frac{p}{\eta}) + 0((S - z - 1) + \cdots + 1)\right]}_{\max_{\sum_{j=1}^{S} s_j = e,\, 0 \le s_j < p/\eta} \sum_{j=1}^{S} j s_j} - \sum_{j=1}^{S} j s_j$$
$$\ge 0$$

and

$$Opt\big(I(\sigma)\big) - Opt\big(I(\sigma')\big)$$
$$= \frac{1}{2}\frac{p}{\eta}(S(S+1) - S'(S'+1)) + \frac{p}{\eta}(SM - S'M') + \frac{1}{2}[T_M(M+1) - T'_M(M'+1)]$$
$$\quad + (\frac{p}{\eta}(S - S') + T_M - T'_M)L$$
$$= \frac{pz}{2\eta}(2S - z + 1) + \frac{pz}{\eta}(z + M - S) - \frac{z}{2}\left[T_M + M\frac{p}{\eta} + z\frac{p}{\eta} + \frac{p}{\eta}\right]$$
$$= \frac{pz}{2\eta}(z + 2M + 1) - \frac{zT_M}{2} - \frac{pz}{2\eta}(M + z + 1)$$
$$= \frac{z}{2}(\frac{p}{\eta}M - T_M)$$
$$\le 0 \quad \textit{(every intermediate job has size at least } \frac{p}{\eta}\textit{, so } \frac{p}{\eta}M \le T_M\textit{.)} \qquad \square$$

*Proof for Lemma B.4.* It follows from LemmasB.5 and B.6. $\qquad\square$

We now refine the ranges for $x, y, e$. Before $t^* = (\eta p)\beta^* n$, the scheduler $A$ can complete at most $\beta^* n$ large jobs, so $x \leq \beta^*$. There is an upper bound of $1 - \beta^*$ on $y$, since if $yn \geq (1-\beta^*)n+1$, the total processing of jobs prior to $t^*$ is at most $(n - ((1-\beta^*)n+1)) \cdot (\eta p) + e = (\beta^* n - 1)(\eta p) + e < (\eta p)\beta^* n = t^*$. This contradicts with that scheduler $A$ has processed $t^*$ amount of time on jobs. Finally, by the setting of adversary, we have $\frac{T_M}{M} \geq \frac{p}{\eta} \Leftrightarrow \frac{t^* - (\eta p)xn - e}{(1-x-y)n} \geq \frac{p}{\eta} \Leftrightarrow y \geq \eta^2 x + (1-x) - \frac{\eta^2}{1+\eta} + \frac{e\eta}{pn}$. Thus, we formulate the feasible range for the triple:

$$0 \leq x \leq \frac{1}{\eta+1}, \ \max\{0, \eta^2 x + (1-x) - \frac{\eta^2}{1+\eta} + \frac{e\eta}{pn}\} \leq y \leq \frac{\eta}{\eta+1}, \ 0 \leq e < \frac{p}{\eta}$$

Let $A$ be the online algorithm that minimizes the competitive ratio against the adversary. By Lemma B.4, the schedule can be determined by three parameters $x, y, e$. We can write $A(I)$ and $Opt(I)$ under any given choice of $(x, y, e)$, denoted by $A_{(x,y,e)}(I)$ and $Opt_{(x,y,e)}(I)$. Let $L = xn$, $S = yn$, $M = (1-x-y)n$, $l = \eta p$, $s = \frac{p}{\eta}$, and $m = \frac{(t^* - (\eta p)xn - e)}{(1-x-y)n}$. We write:

$$A_{(x,y,e)}(I)$$
$$= l \cdot \underbrace{\frac{L(L+1)}{2}}_{\text{large}} + \underbrace{t^* M}_{\text{intermediate}} + \underbrace{t^* S + s \cdot \frac{S(S+1)}{2} - eS}_{\text{small}}$$

$$= \underbrace{\eta p \frac{(xn)(xn+1)}{2}}_{\text{large}} + \underbrace{t^* (1-x-y)n}_{\text{intermediate}} + \underbrace{t^*(yn) + \frac{p}{\eta} \frac{(yn)(yn+1)}{2} - e(yn)}_{\text{small}}$$

$$= \underbrace{\eta p\left(\frac{x^2 n^2}{2} + \frac{xn}{2}\right)}_{\text{large}} + \underbrace{\beta^* \eta p (1-x-y) n^2}_{\text{intermediate}} + \underbrace{\beta^* \eta p\, y\, n^2 + \frac{p}{\eta}\left(\frac{y^2 n^2}{2} + \frac{yn}{2}\right) - e\, y\, n}_{\text{small}}$$

$$= (\frac{1}{2}\eta p x^2 + \beta^* \eta p(1-x) + \frac{1}{2\eta}py^2) \cdot n^2 + (\frac{1}{2}\eta p x + \frac{1}{2\eta}py - ey) \cdot n$$

and

$$Opt_{(x,y,e)}(I)$$
$$= s \cdot \underbrace{\frac{S(S+1)}{2}}_{\text{small}} + \underbrace{sSM + m \cdot \frac{M(M+1)}{2}}_{\text{intermediate}} + \underbrace{(sS + mM)L + l \cdot \frac{L(L+1)}{2}}_{\text{large}}$$

$$= \underbrace{s \frac{(yn)(yn+1)}{2}}_{\text{small}} + \underbrace{s(yn)((1-x-y)n) + m \frac{((1-x-y)n)((1-x-y)n+1)}{2}}_{\text{intermediate}}$$

$$+ \underbrace{(s(yn) + m((1-x-y)n))(xn) + l \frac{(xn)(xn+1)}{2}}_{\text{large}}$$

$$= \left[\frac{p}{\eta} \frac{y^2 n^2 + yn}{2} + \frac{p}{\eta} y(1-x-y) n^2 + (\eta p(\beta^* - x)n - e)\frac{(1-x-y)n+1}{2}\right]$$

$$+ \left[(\frac{p}{\eta}y\,n + \eta p(\beta^* - x)n - e)\,x\,n + \eta p \frac{x^2 n^2 + xn}{2}\right]$$

$$= \left(\frac{1}{2\eta}py(2-y) + \frac{1}{2}\eta p[x^2 + (\beta^* - x)(1+x-y)]\right) n^2 + (\frac{1}{2}\eta p\beta^* + \frac{1}{2\eta}py - \frac{1}{2}e(1+x-y))n$$

$$- \frac{1}{2}e$$

We consider the following optimization:

$$\min_{x,y,e} \frac{A_{(x,y,e)}(I)}{Opt_{(x,y,e)}(I)} \tag{3}$$

$$\text{s.t.} 0 \le x \le \frac{1}{1+\eta}, \max\left\{0, \eta^2 x + (1-x) - \frac{\eta^2}{\eta+1} + \frac{e\eta}{pn}\right\} \le y \le \frac{\eta}{\eta+1}, 0 \le e < \frac{p}{\eta}$$

We will show the following bound, which will prove Theorem 4.1.

**Lemma B.7.** *For Optimization (3), we have:*

$$\lim_{n\to\infty} \frac{A_{(x,y,e)}(I)}{Opt_{(x,y,e)}(I)} \ge \begin{cases} \eta, & \eta \le \eta^* \\ \lambda(\eta), & \eta^* < \eta \le 1 + \sqrt{3} \\ 2, & \eta > 1 + \sqrt{3} \end{cases}$$

*for any feasible $(x, y, e)$, where $\eta^*$ is the positive root of $x^4 - 4x - 4 = 0$ ($\eta^* \approx 1.835$), and*
$$\lambda(\eta) = \frac{2\eta\left(\eta^2 + \eta + \sqrt{(\eta+1)(2\eta^4 + \eta^3 + 9\eta^2 + 16\eta + 8)}\right)}{\eta^4 + 4\eta^2 + 8\eta + 4}.$$

Our proof structure is as follows. Observe that terms $A_{(x,y,e)}(I)$ and $Opt_{(x,y,e)}(I)$ are quadratic in $n$ and the coefficients of $n^2$ are positive for any feasible $(x, y, e)$. The adversary can set sufficiently large $n$ so $\lim_{n\to\infty} \frac{A_{(x,y,e)}(I)}{Opt_{(x,y,e)}(I)}$ equals the ratio of the coefficients of $n^2$ in the numerator and denominator. Write the coefficient of $n^2$ in $A_{(x,y,e)}(I) - t \cdot Opt_{(x,y,e)}(I)$:

$$\left(\frac{1}{2}\eta p x^2 + \beta^* \eta p(1-x) + \frac{1}{2\eta} py^2\right) - t \cdot \left(\frac{1}{2\eta}py(2-y) + \frac{1}{2}\eta p[x^2 + (\beta^* - x)(1+x-y)]\right)$$
$$= \frac{p}{2\eta(\eta+1)} \cdot [y^2(\eta t + \eta + t + 1) + y(-\eta^3 tx - \eta^2 tx + \eta^2 t - 2\eta t - 2t)$$
$$+ (\eta^3 tx + \eta^3 x^2 - \eta^2 t + \eta^2 x^2 - 2\eta^2 x + 2\eta^2)] \qquad \text{(substituting } \beta^* = 1/(\eta+1))$$
$$= \frac{p}{2\eta(\eta+1)} \cdot h(x, y, t)$$

where $h(x, y, t) = y^2(\eta t + \eta + t + 1) + y(-\eta^3 tx - \eta^2 tx + \eta^2 t - 2\eta t - 2t) + (\eta^3 tx + \eta^3 x^2 - \eta^2 t + \eta^2 x^2 - 2\eta^2 x + 2\eta^2)$. Therefore, showing

$$\lim_{n\to\infty} \frac{A_{(x,y,e)}(I)}{Opt_{(x,y,e)}(I)} \ge t$$

for some $t$ is equivalent to showing

$$h(x, y, t) \ge 0$$

Notice that the expression depends on $x$ and $y$ but not $e$, which is intuitive as $e$ only affects a small portion of processor allocation. Observe that $h$ is quadratic in $x$ and $y$, if the other variables are fixed as constants. Therefore, with any fixed $x$, the minimum of $h$ occurs at either the boundaries of $y$ or at $\tilde{y} = \frac{t(\eta^3 x + \eta^2 x - \eta^2 + 2\eta + 2)}{2(\eta+1)(t+1)}$. Similarly, with any fixed $y$, the minimum of $h$ occurs at either the boundaries of $x$ or $\tilde{x} = \frac{\eta ty - \eta t + ty + 2}{2(\eta+1)}$. To see this, write $h(x, y, t) = x^2(\eta^3 + \eta^2) + x(-\eta^3 ty + \eta^3 t - \eta^2 ty - 2\eta^2) + (\eta^2 ty - \eta^2 t + 2\eta^2 + \eta ty^2 - 2\eta ty + \eta y^2 + ty^2 - 2ty + y^2)$. Given these candidates for the minimum, we can show Lemma B.7 by discussing three cases.

**Lemma B.8.** *For Optimization (3), we have:*

$$\lim_{n\to\infty} \frac{A_{(x,y,e)}(I)}{Opt_{(x,y,e)}(I)} \ge \eta, \ \eta \le \eta^*$$

*for any feasible $(x, y, e)$, where $\eta^*$ is the positive root of $x^4 - 4x - 4 = 0$ ($\eta^* \approx 1.835$).*

*Proof.* We show that the claim holds for $0 \le x \le \frac{1}{1+\eta}, 0 \le y \le \frac{\eta}{\eta+1}, 0 \le e < \frac{p}{\eta}$, which relaxes the range for $y$. Thus, it will hold for a more restricted region. Write:

$$h(x, y, \eta) = y^2(\eta+1)^2 - y \cdot \eta\left(\eta^3 x + \eta^2 x - \eta^2 + 2\eta + 2\right) + \eta^2\left(\eta^2 x + \eta x^2 - \eta + x^2 - 2x + 2\right)$$

We show that $h(x, y, \eta) \ge 0$ where the equality holds when $(x, y) = (\frac{1}{\eta+1}, \frac{\eta}{\eta+1})$. Notice that $x = \frac{1}{\eta+1} \Rightarrow y = \frac{\eta}{\eta+1}, e = 0$, as $x = \frac{1}{\eta+1}$ implies that the processing must be all allocated to the

large jobs by time $t^*$, so there will be no intermediate jobs and $0$ time allocated to small jobs by time $t^*$. Given any $x$, we have $h(x, y, \eta) \geq \min\{h(x, 0, \eta), h(x, \frac{\eta}{\eta+1}, \eta), h(x, \tilde{y}, \eta)\}$. Given any $y$, we have $h(x, y, \eta) \geq \min\{h(0, y, \eta), h(\frac{1}{\eta+1}, y, \eta), h(\tilde{x}, y, \eta)\}$.

**Case 1** ($y = 0$). We have:

$$h(x, 0, \eta) = \eta^2 \left(\eta^2 x + \eta x^2 - \eta + x^2 - 2x + 2\right) = \eta^2 \left((\eta+1)x^2 + (\eta^2 - 2)x + 2 - \eta\right)$$

**Case 1.1** ($y = 0$, $x = 0$). We have:

$$h(0, 0, \eta) = \eta^2(2 - \eta) > 0$$

for $\eta \leq \eta^*$ ($\eta^* \approx 1.835$).

**Case 1.2** ($y = 0$, $x = \frac{1}{\eta+1}$). We have:

$$h(\frac{1}{\eta+1}, 0, \eta) = \eta^2 \left((\eta+1)(\frac{1}{\eta+1})^2 + (\eta^2 - 2)(\frac{1}{\eta+1}) + 2 - \eta\right) = \eta^2 > 0$$

**Case 1.3** ($y = 0$, $x = \tilde{x} = \frac{2-\eta^2}{2(\eta+1)}$). We have:

$$h(\frac{2-\eta^2}{2(\eta+1)}, 0, \eta) = \eta^2 \left((\eta+1)(\frac{2-\eta^2}{2(\eta+1)})^2 + (\eta^2 - 2)\frac{2-\eta^2}{2(\eta+1)} + 2 - \eta\right)$$

$$= -\frac{\eta^2 \left(\eta^4 - 4\eta - 4\right)}{4(\eta+1)} \geq 0$$

if $\eta \leq \eta^*$, which implies $\eta^4 - 4\eta - 4 \leq 0$.

Putting Cases 1.1, 1.2, and 1.3 together, we have:

$$h(x, 0, \eta) \geq \min\{h(0, 0, \eta), h(\frac{1}{\eta+1}, 0, \eta), h(\tilde{x}, 0, \eta)\} \geq 0$$

**Case 2** ($y = \frac{\eta}{\eta+1}$). We have:

$$h(x, \frac{\eta}{\eta+1}, \eta) = (\frac{\eta}{\eta+1})^2(\eta+1)^2 - \frac{\eta}{\eta+1} \cdot \eta \left(\eta^3 x + \eta^2 x - \eta^2 + 2\eta + 2\right)$$

$$+ \eta^2 \left(\eta^2 x + \eta x^2 - \eta + x^2 - 2x + 2\right)$$

$$= (\eta+1)(x - \frac{1}{\eta+1})^2 \geq 0$$

where equality holds when $(x, y) = (\frac{1}{\eta+1}, \frac{\eta}{\eta+1})$.

**Case 3** ($y = \tilde{y} = \frac{\eta(\eta^3 x + \eta^2 x - \eta^2 + 2\eta + 2)}{2(\eta+1)(\eta+1)} = \frac{\eta(\eta^3 x + \eta^2 x - \eta^2 + 2\eta + 2)}{2(\eta+1)^2}$). We have:

$$h(x, \tilde{y}, \eta) = (\frac{\eta \left(\eta^3 x + \eta^2 x - \eta^2 + 2\eta + 2\right)}{2(\eta+1)^2})^2(\eta+1)^2$$

$$- \frac{\eta \left(\eta^3 x + \eta^2 x - \eta^2 + 2\eta + 2\right)}{2(\eta+1)^2} \cdot \eta \left(\eta^3 x + \eta^2 x - \eta^2 + 2\eta + 2\right)$$

$$+ \eta^2 \left(\eta^2 x + \eta x^2 - \eta + x^2 - 2x + 2\right)$$

$$= -\frac{\eta^2}{4}(\eta^4 - 4\eta - 4)(x - \frac{1}{\eta+1})^2 \geq 0$$

where equality holds when $(x, y) = (\frac{1}{\eta+1}, \frac{\eta}{\eta+1})$.

Putting Cases 1, 2, and 3, together, we have:

$$h(x, y, \eta) \geq \min\{h(x, 0, \eta), h(x, \frac{\eta}{\eta+1}, \eta), h(x, \tilde{y}, \eta)\} \geq 0, \; \forall \, \eta \leq \eta^* \qquad \square$$

Before we show the case for $\eta^* < \eta \le 1 + \sqrt{3}$, we need a few bounds for $\lambda(\eta)$.

**Lemma B.9.**
$$\eta^* \le \lambda(\eta) \le 2$$

*for $\eta^* < \eta \le 1 + \sqrt{3}$, where $\eta^*$ is the positive root of $x^4 - 4x - 4 = 0$ ($\eta^* \approx 1.835$) and $\lambda(\eta) = \frac{2\eta\left(\eta^2 + \eta + \sqrt{(\eta+1)(2\eta^4 + \eta^3 + 9\eta^2 + 16\eta + 8)}\right)}{\eta^4 + 4\eta^2 + 8\eta + 4}$.*

*Proof.* We first show that $\lambda(\eta)$ is increasing in $[\eta^*, 1 + \sqrt{3}]$. Differentiate $\lambda$ with respect to $\eta$ and factorize:

$$\frac{d}{d\eta}\lambda(\eta) = -\frac{2(\eta + 2)(\eta^2 - 2\eta - 2)(\eta^2 + 2\eta + 2)N(\eta)}{\sqrt{\eta + 1}(\eta^4 + 4\eta^2 + 8\eta + 4)^2\sqrt{2\eta^4 + \eta^3 + 9\eta^2 + 16\eta + 8}}$$

where:

$$N(\eta) = \eta^4 + \eta^3 + 5\eta^2 + \eta\sqrt{2\eta^5 + 3\eta^4 + 10\eta^3 + 25\eta^2 + 24\eta + 8} + 8\eta + 4$$

The sign of $\frac{d}{d\eta}\lambda(\eta)$ is determined by $-(\eta^2 - 2\eta - 2) = (x + \sqrt{3} - 1)((1 + \sqrt{3}) - x)$. Therefore,

$$\frac{d}{d\eta}\lambda(\eta) \ge 0, \ \eta^* < \eta \le 1 + \sqrt{3}$$

and, thus, $\lambda(\eta)$ is increasing in $[\eta^*, 1 + \sqrt{3}]$. Then, to show the claim it suffices to show that $\lambda(\eta) = \eta$ if $\eta = \eta^*$ and $\lambda(\eta) = 2$ if $\eta = 1 + \sqrt{3}$.

**Case 1** ($\lambda(\eta) = \eta$ **if** $\eta = \eta^*$)**.** Let $\eta^* > 0$ satisfy $(\eta^*)^4 = 4\eta^* + 4$. Substitute $\eta = \eta^*$ and use $(\eta^*)^4 = 4\eta^* + 4$. We have:

$$2(\eta^*)^4 + (\eta^*)^3 + 9(\eta^*)^2 + 16\eta^* + 8 = 2(4\eta^* + 4) + (\eta^*)^3 + 9(\eta^*)^2 + 16\eta^* + 8$$
$$= (\eta^* + 1)(\eta^* + 4)^2$$

Therefore:

$$\lambda(\eta^*) = \frac{2\eta^*\left(\eta^{*2} + \eta^* + (\eta^* + 1)(\eta^* + 4)\right)}{4(\eta^* + 1)(\eta^* + 2)} = \frac{4\eta^*(\eta^* + 1)(\eta^* + 2)}{4(\eta^* + 1)(\eta^* + 2)} = \eta^*$$

**Case 2** ($\lambda(\eta) = 2$ **if** $\eta = 1 + \sqrt{3}$)**.** We have $\eta^2 = 2\eta + 2$. The denominator of $\lambda$ becomes

$$\eta^4 + 4\eta^2 + 8\eta + 4 = (2\eta + 2)^2 + 4\eta^2 + 8\eta + 4 = 8(\eta + 1)^2.$$

Next, compute the radicand in the numerator:

$$2\eta^4 + \eta^3 + 9\eta^2 + 16\eta + 8 = 2(2\eta + 2)^2 + \eta(2\eta + 2) + 9\eta^2 + 16\eta + 8$$
$$= 8\eta^2 + 16\eta + 8 + (2\eta^2 + 2\eta) + 9\eta^2 + 16\eta + 8$$
$$= 19\eta^2 + 34\eta + 16$$
$$= 18\eta^2 + (2\eta + 2) + 34\eta + 16 = 18(\eta + 1)^2$$

We have:

$$\lambda(\eta) = \frac{2\eta\left(\eta^2 + \eta + 3(\eta + 1)\sqrt{2(\eta + 1)}\right)}{8(\eta + 1)^2} = \frac{2\eta\left(\eta^2 + \eta + 3(\eta + 1)\eta\right)}{8(\eta + 1)^2} = \frac{\eta^2}{\eta + 1} = \frac{\eta^2}{\eta^2/2} = 2$$

$\square$

**Lemma B.10.**
$$\lambda(\eta) \le \eta$$

*for $\eta^* < \eta \le 1 + \sqrt{3}$, where $\eta^*$ is the positive root of $x^4 - 4x - 4 = 0$ ($\eta^* \approx 1.835$) and $\lambda(\eta) = \frac{2\eta\left(\eta^2 + \eta + \sqrt{(\eta+1)(2\eta^4 + \eta^3 + 9\eta^2 + 16\eta + 8)}\right)}{\eta^4 + 4\eta^2 + 8\eta + 4}$.*

*Proof.* To show

$$\frac{2\eta \left( \eta^2 + \eta + \sqrt{(\eta+1)(2\eta^4 + \eta^3 + 9\eta^2 + 16\eta + 8)} \right)}{\eta^4 + 4\eta^2 + 8\eta + 4} \leq \eta$$

It suffices to show

$$2\sqrt{(\eta+1)(2\eta^4 + \eta^3 + 9\eta^2 + 16\eta + 8)} \leq (\eta^4 + 4\eta^2 + 8\eta + 4) - 2(\eta^2 + \eta)$$
$$\Leftrightarrow 4(\eta+1)(2\eta^4 + \eta^3 + 9\eta^2 + 16\eta + 8) \leq (\eta^4 + 2\eta^2 + 6\eta + 4)^2$$
$$\Leftrightarrow -\left(\eta^4 - 4\eta - 4\right)\left(\eta^4 + 4\eta^2 + 8\eta + 4\right) \leq 0$$

which is trivial as $\eta^* < \eta$ ($\Rightarrow \eta^4 - 4\eta - 4 > 0$). $\qquad\square$

**Lemma B.11.** *For Optimization (3), we have:*

$$\lim_{n\to\infty} \frac{A_{(x,y,e)}(I)}{Opt_{(x,y,e)}(I)} \geq \lambda(\eta), \ \eta^* < \eta \leq 1 + \sqrt{3}$$

*for any feasible $(x, y, e)$, where $\eta^*$ is the positive root of $x^4 - 4x - 4 = 0$ ($\eta^* \approx 1.835$), and*
$\lambda(\eta) = \frac{2\eta\left(\eta^2 + \eta + \sqrt{(\eta+1)(2\eta^4 + \eta^3 + 9\eta^2 + 16\eta + 8)}\right)}{\eta^4 + 4\eta^2 + 8\eta + 4}$.

*Proof.* We show that the claim holds for $0 \leq x \leq \frac{1}{1+\eta}, 0 \leq y \leq \frac{\eta}{\eta+1}, 0 \leq e < \frac{p}{\eta}$, which relaxes the range for $y$. Thus, it will hold for a more restricted region. Recall that $h(x, y, t) = x^2 \left(\eta^3 + \eta^2\right) + x \left(-\eta^3 ty + \eta^3 t - \eta^2 ty - 2\eta^2\right) + (\eta^2 ty - \eta^2 t + 2\eta^2 + \eta ty^2 - 2\eta ty + \eta y^2 + ty^2 - 2ty + y^2)$. We show that $h(x, y, \lambda(\eta)) \geq 0$ where the equality holds when $(x, y) = (0, \frac{\lambda(\eta)(-\eta^2 + 2\eta + 2)}{2(\eta+1)(\lambda(\eta)+1)})$. Given any $x$, we have $h(x, y, t) \geq \min\{h(x, 0, t), h(x, \frac{\eta}{\eta+1}, t), h(x, \tilde{y}, t)\}$. Given any $y$, we have $h(x, y, t) \geq \min\{h(0, y, t), h(\frac{1}{\eta+1}, y, t), h(\tilde{x}, y, t)\}$.

**Case 1 ($y = 0$).** We have:

$$h(x, 0, t)$$
$$= x^2 \left(\eta^3 + \eta^2\right) + x \left(\eta^3 t - 2\eta^2\right) + (-\eta^2 t + 2\eta^2)$$
$$= \eta^2((\eta+1)x^2 + (\eta t - 2)x + (2 - t))$$

Using Lemma B.9, we have $\lambda(\eta) \leq 2 \Rightarrow 2 - \lambda(\eta) \geq 2$ and $\eta^* < \eta, \eta^* \leq \lambda(\eta) \Rightarrow \eta\lambda(\eta) > (\eta^*)^2 > 2$. Therefore,

$$h(x, 0, \lambda(\eta)) = \eta^2((\eta+1)x^2 + (\eta\lambda(\eta) - 2)x + (2 - \lambda(\eta))) > 0$$

**Case 2 ($y = \frac{\eta}{\eta+1}$).** We have:

$$h(x, \frac{\eta}{\eta+1}, t) = \left(\eta^3 + 2\eta^2 + \eta\right)x^2 + \left(-2\eta^2 - 2\eta\right)x + 2\eta^2 - 2\eta t + 3\eta - 2t$$

**Case 2.1 ($y = \frac{\eta}{\eta+1}, x = 0$).** We have:

$$h(0, \frac{\eta}{\eta+1}, \lambda(\eta)) = 2\eta^2 - 2\eta\lambda(\eta) + 3\eta - 2\lambda(\eta) \geq 2\eta^2 - 2\eta^2 + 3\eta - 2\eta = \eta > 0$$

where $\lambda(\eta) \leq \eta$ is by Lemma B.10.

If $y = \frac{\eta}{\eta+1}$, we have $\tilde{x} = \frac{\eta ty - \eta t + ty + 2}{2(\eta+1)} = \frac{\eta t \frac{\eta}{\eta+1} - \eta t + t \frac{\eta}{\eta+1} + 2}{2(\eta+1)} = \frac{1}{\eta+1}$. Therefore, we are left with one case to consider under $y = \frac{\eta}{\eta+1}$.

**Case 2.2 ($y = \frac{\eta}{\eta+1}, x = \frac{1}{\eta+1}$).** We have:

$$h(\frac{1}{\eta+1}, \frac{\eta}{\eta+1}, t) = \left(\eta^3 + 2\eta^2 + \eta\right)(\frac{1}{\eta+1})^2 + \left(-2\eta^2 - 2\eta\right)\frac{1}{\eta+1} + 2\eta^2 - 2\eta t + 3\eta - 2t$$

$$= 2\eta(\eta - t)$$

Therefore,

$$h(\frac{1}{\eta+1}, \frac{\eta}{\eta+1}, \lambda(\eta)) = 2\eta(\eta - \lambda(\eta)) \geq 0$$

for $\eta^* < \eta$.

Putting Cases 2.1 and 2.2 together, we have:

$$h(x, \tfrac{\eta}{\eta+1}, \lambda(\eta)) \geq \min\{h(0, \tfrac{\eta}{\eta+1}, \lambda(\eta)), h(\tfrac{1}{\eta+1}, \tfrac{\eta}{\eta+1}, \lambda(\eta)), h(\tilde{x}, \tfrac{\eta}{\eta+1}, \lambda(\eta))\} \geq 0$$

**Case 3** ($y = \tilde{y} = \frac{t(\eta^3 x + \eta^2 x - \eta^2 + 2\eta + 2)}{2(\eta+1)(t+1)}, t = \lambda(\eta)$)**. We have:**

$$h(x, \tilde{y}, t) = (\frac{t\left(\eta^3 x + \eta^2 x - \eta^2 + 2\eta + 2\right)}{2(\eta+1)(t+1)})^2(\eta+1)^2$$
$$- \frac{t\left(\eta^3 x + \eta^2 x - \eta^2 + 2\eta + 2\right)}{2(\eta+1)(t+1)} \cdot \eta\left(\eta^3 x + \eta^2 x - \eta^2 + 2\eta + 2\right)$$
$$+ \eta^2\left(\eta^2 x + \eta x^2 - \eta + x^2 - 2x + 2\right)$$

**Case 3.1** ($y = \tilde{y} = \frac{t(\eta^3 x + \eta^2 x - \eta^2 + 2\eta + 2)}{2(\eta+1)(t+1)}, t = \lambda(\eta), x = 0$)**. We have:**

$$h(0, \tilde{y}, t)$$
$$= (\frac{t\left(-\eta^2 + 2\eta + 2\right)}{2(\eta+1)(t+1)})^2(\eta+1)^2 - \frac{t\left(-\eta^2 + 2\eta + 2\right)}{2(\eta+1)(t+1)} \cdot \eta\left(-\eta^2 + 2\eta + 2\right) + \eta^2\left(-\eta + 2\right)$$
$$= \frac{\left(-\eta^4 - 4\eta^2 - 8\eta - 4\right)t^2 + \left(4\eta^3 + 4\eta^2\right)t + 8\eta^3 + 8\eta^2}{4\left(\eta+1\right)\left(t+1\right)} = 0$$

The equality holds because $\lambda(\eta)$ is the positive solution to equation $\left(-\eta^4 - 4\eta^2 - 8\eta - 4\right)x^2 + \left(4\eta^3 + 4\eta^2\right)x + 8\eta^3 + 8\eta^2 = 0$. This can be verified via the quadratic formula:

$$\frac{-\left(4\eta^3 + 4\eta^2\right) - \sqrt{\left(4\eta^3 + 4\eta^2\right)^2 - 4\left(-\eta^4 - 4\eta^2 - 8\eta - 4\right)\left(8\eta^3 + 8\eta^2\right)}}{2\left(-\eta^4 - 4\eta^2 - 8\eta - 4\right)}$$
$$= \frac{\left(4\eta^3 + 4\eta^2\right) + \sqrt{16\eta^2\left(\eta+1\right)\left(2\eta^4 + \eta^3 + 9\eta^2 + 16\eta + 8\right)}}{2\left(\eta^4 + 4\eta^2 + 8\eta + 4\right)}$$
$$= \frac{2\eta\left(\eta^2 + \eta + \sqrt{(\eta+1)(2\eta^4 + \eta^3 + 9\eta^2 + 16\eta + 8)}\right)}{\eta^4 + 4\eta^2 + 8\eta + 4} = \lambda(\eta)$$

**Case 3.2** ($y = \tilde{y} = \frac{t(\eta^3 x + \eta^2 x - \eta^2 + 2\eta + 2)}{2(\eta+1)(t+1)}, t = \lambda(\eta), x = \frac{1}{\eta+1}$)**. We have:**

$$h(\frac{1}{\eta+1}, \tilde{y}, t) = (\frac{t\left(\eta^3 \frac{1}{\eta+1} + \eta^2 \frac{1}{\eta+1} - \eta^2 + 2\eta + 2\right)}{2(\eta+1)(t+1)})^2(\eta+1)^2$$
$$- \frac{t\left(\eta^3 \frac{1}{\eta+1} + \eta^2 \frac{1}{\eta+1} - \eta^2 + 2\eta + 2\right)}{2(\eta+1)(t+1)} \cdot \eta\left(\eta^3 \frac{1}{\eta+1} + \eta^2 \frac{1}{\eta+1} - \eta^2 + 2\eta + 2\right)$$
$$+ \eta^2\left(\eta^2 \frac{1}{\eta+1} + \eta(\frac{1}{\eta+1})^2 - \eta + (\frac{1}{\eta+1})^2 - 2\frac{1}{\eta+1} + 2\right)$$
$$= \frac{(\eta - t)\left(2\eta^2 t + 2\eta^2 + 2\eta t + \eta + t\right)}{(\eta+1)(t+1)}$$

Therefore,

$$h(\frac{1}{\eta+1}, \tilde{y}, \lambda(\eta)) = \frac{(\eta - \lambda(\eta))\left(2\eta^2\lambda(\eta) + 2\eta^2 + 2\eta\lambda(\eta) + \eta + \lambda(\eta)\right)}{(\eta+1)(t+1)} \geq 0$$

for $\lambda(\eta) \leq \eta$ by Lemma B.10.

We then show that under $y = \tilde{y}$, $x = \tilde{x} \geq \frac{1}{\eta+1}$, so this case does not need to be handled.

**Case 3.3** ($y = \tilde{y} = \frac{t(\eta^3 x + \eta^2 x - \eta^2 + 2\eta + 2)}{2(\eta+1)(t+1)}$, $t = \lambda(\eta)$, $x = \tilde{x}$).

We first derive the expression for $\tilde{x}$ as a function of $\eta$ and $t$.

$$\tilde{x}$$
$$= \frac{\eta t \tilde{y} - \eta t + t \tilde{y} + 2}{2(\eta+1)}$$
$$= \frac{1}{2(\eta+1)} \left[ (\eta t + t)\tilde{y} + 2 - \eta t \right]$$
$$= \frac{1}{2(\eta+1)} \left[ (\eta t + t)\frac{t(\eta^3 \tilde{x} + \eta^2 \tilde{x} - \eta^2 + 2\eta + 2)}{2(\eta+1)(t+1)} + 2 - \eta t \right]$$

Solving for $\tilde{x}$ gives:

$$\tilde{x} = \frac{t^2(2 - \eta^2) + 2(2 - \eta)t + 4}{(\eta+1)\left[4(t+1) - \eta^2 t^2\right]}$$

However,

$$\tilde{x} - \frac{1}{\eta+1}$$
$$= \frac{t^2(2 - \eta^2) + 2(2 - \eta)t + 4}{(\eta+1)\left[4(t+1) - \eta^2 t^2\right]} - \frac{1}{\eta+1}$$
$$= \frac{2t(\eta - t)}{(\eta+1)(\eta^2 t^2 - 4t - 4)}$$

Observe that $\eta - \lambda(\eta) \geq 0$ and

$$\eta^2(\lambda(\eta))^2 - 4\lambda(\eta) - 4 \geq (\lambda(\eta))^4 - 4\lambda(\eta) - 4 \geq 0$$

where the first inequality is due to Lemma B.10 and the second is due to $\lambda(\eta) \geq \eta^*$ (Lemma B.10). Therefore,

$$\tilde{x} \geq \frac{1}{\eta+1}$$

The case does not exist or has already been taken into account in Case 3.2.

Putting Case 3.1 and 3.2 together, we have:

$$h(x, \tilde{y}, \lambda(\eta)) \geq \min\{h(0, \tilde{y}, \lambda(\eta)), h(\tfrac{1}{\eta+1}, \tilde{y}, \lambda(\eta))\} \geq 0 \qquad \qquad \square$$

**Lemma B.12.** *For Optimization (3), we have:*

$$\lim_{n \to \infty} \frac{A_{(x,y,e)}(I)}{Opt_{(x,y,e)}(I)} \geq 2, \ \eta > 1 + \sqrt{3}$$

*for any feasible $(x, y, e)$.*

*Proof.* We show that the claim holds for $0 \leq x \leq \frac{1}{1+\eta}, 0 \leq y \leq \frac{\eta}{\eta+1}, 0 \leq e < \frac{p}{\eta}$, which relaxes the range for $y$. Thus, it will hold for a more restricted region. Recall that $h(x, y, t) = x^2\left(\eta^3 + \eta^2\right) + x\left(-\eta^3 ty + \eta^3 t - \eta^2 ty - 2\eta^2\right) + \left(\eta^2 ty - \eta^2 t + 2\eta^2 + \eta ty^2 - 2\eta ty + \eta y^2 + ty^2 - 2ty + y^2\right)$. We show that $h(x, y, 2) \geq 0$ where the equality holds when $(x, y) = (0, 0)$. Given any $x$, we have $h(x, y, t) \geq \min\{h(x, 0, t), h(x, \frac{\eta}{\eta+1}, t), h(x, \tilde{y}, t)\}$. Given any $y$, we have $h(x, y, t) \geq \min\{h(0, y, t), h(\frac{1}{\eta+1}, y, t), h(\tilde{x}, y, t)\}$.

**Case 1** ($y = 0$). We have:

$$h(x, 0, t) = \eta^2((\eta+1)x^2 + (\eta t - 2)x + (2 - t))$$

Therefore,

$$h(x, 0, 2) = \eta^2((\eta + 1)x^2 + 2(\eta - 1)x) \geq 0$$

**Case 2 ($y = \frac{\eta}{\eta+1}$).** We have:

$$h(x, \frac{\eta}{\eta + 1}, 2) = \left(\eta^3 + 2\eta^2 + \eta\right) x^2 + \left(-2\eta^2 - 2\eta\right) x + 2\eta^2 - \eta - 4$$

**Case 2.1 ($y = \frac{\eta}{\eta+1}$, $x = 0$).** We have:

$$h(0, \frac{\eta}{\eta + 1}, 2) = 2\eta^2 - \eta - 4 > 0$$

for $\eta > 1 + \sqrt{3}$. If $y = \frac{\eta}{\eta+1}$, we have $\tilde{x} = \frac{1}{\eta+1}$, so we have one case left under $y = \frac{\eta}{\eta+1}$.

**Case 2.2 ($y = \frac{\eta}{\eta+1}$, $x = \frac{1}{\eta+1}$).** We have:

$$h(\frac{1}{\eta + 1}, \frac{\eta}{\eta + 1}, 2) = 2\eta(\eta - 2) \geq 0$$

for $\eta > 1 + \sqrt{3}$. Putting Cases 2.1 and 2.2 together, we have:

$$h(x, \frac{\eta}{\eta+1}, 2) \geq \min\{h(0, \frac{\eta}{\eta+1}, 2), h(\frac{1}{\eta+1}, \frac{\eta}{\eta+1}, 2), h(\tilde{x}, \frac{\eta}{\eta+1}, 2)\} \geq 0$$

**Case 3 ($y = \tilde{y} = \frac{\left(\eta^3 x + \eta^2 x - \eta^2 + 2\eta + 2\right)}{3(\eta+1)}$).**

$$
\begin{aligned}
h(x, \tilde{y}, 2) = {} & (\frac{\left(\eta^3 x + \eta^2 x - \eta^2 + 2\eta + 2\right)}{3(\eta + 1)})^2 (\eta + 1)^2 \\
& - \frac{\left(\eta^3 x + \eta^2 x - \eta^2 + 2\eta + 2\right)}{3(\eta + 1)} \cdot \eta \left(\eta^3 x + \eta^2 x - \eta^2 + 2\eta + 2\right) \\
& + \eta^2 \left(\eta^2 x + \eta x^2 - \eta + x^2 - 2x + 2\right)
\end{aligned}
$$

For this case to exist, we need $\tilde{y} \geq 0$. This implies:

$$\tilde{y} = \frac{\left(\eta^3 x + \eta^2 x - \eta^2 + 2\eta + 2\right)}{3(\eta + 1)} \geq 0 \Rightarrow x \geq \frac{\eta^2 - 2\eta - 2}{\eta^2(\eta + 1)}$$

Therefore, the range of $x$ becomes $[\frac{\eta^2-2\eta-2}{\eta^2(\eta+1)}, \frac{1}{\eta+1}]$.

**Case 3.1 ($y = \tilde{y} = \frac{\left(\eta^3 x + \eta^2 x - \eta^2 + 2\eta + 2\right)}{3(\eta+1)}$, $x = \frac{\eta^2-2\eta-2}{\eta^2(\eta+1)}$)**

$$
\begin{aligned}
& h(\frac{\eta^2-2\eta-2}{\eta^2(\eta+1)}, \tilde{y}, 2) \\
= {} & (\frac{\left((\eta^3 + \eta^2)\frac{\eta^2-2\eta-2}{\eta^2(\eta+1)} - \eta^2 + 2\eta + 2\right)}{3(\eta + 1)})^2 (\eta + 1)^2 \\
& - \frac{\left((\eta^3 + \eta^2)\frac{\eta^2-2\eta-2}{\eta^2(\eta+1)} - \eta^2 + 2\eta + 2\right)}{3(\eta + 1)} \cdot \eta \left((\eta^3 + \eta^2)\frac{\eta^2-2\eta-2}{\eta^2(\eta+1)} - \eta^2 + 2\eta + 2\right) \\
& + \eta^2 \left(\eta^2 \frac{\eta^2-2\eta-2}{\eta^2(\eta+1)} + \eta(\frac{\eta^2-2\eta-2}{\eta^2(\eta+1)})^2 - \eta + (\frac{\eta^2-2\eta-2}{\eta^2(\eta+1)})^2 - 2\frac{\eta^2-2\eta-2}{\eta^2(\eta+1)} + 2\right) \\
= {} & \frac{\left(\eta^2 - 2\eta - 2\right)\left(2\eta^3 - \eta^2 - 2\eta - 2\right)}{\eta^2(\eta + 1)}
\end{aligned}
$$

With $\eta > 1 + \sqrt{3}$, we have $\eta^2 > 2\eta + 2$ and

$$2\eta^3 - \eta^2 - 2\eta - 2 > \eta(2\eta + 2) - \eta^2 - 2\eta - 2 = \eta^2 - 2 > 0$$

for $\eta > 1 + \sqrt{3}$. Therefore, we have:

$$h(\tfrac{\eta^2 - 2\eta - 2}{\eta^2(\eta+1)}, \tilde{y}, 2) \geq 0$$

**Case 3.2** ($y = \tilde{y} = \frac{(\eta^3 x + \eta^2 x - \eta^2 + 2\eta + 2)}{3(\eta+1)}$, $x = \frac{1}{\eta+1}$)

$$
\begin{aligned}
&h(\tfrac{1}{\eta+1}, \tilde{y}, 2) \\
&= (\frac{\left((\eta^3 + \eta^2)\frac{1}{\eta+1} - \eta^2 + 2\eta + 2\right)}{3(\eta+1)})^2(\eta+1)^2 \\
&\quad - \frac{\left((\eta^3 + \eta^2)\frac{1}{\eta+1} - \eta^2 + 2\eta + 2\right)}{3(\eta+1)} \cdot \eta\left((\eta^3 + \eta^2)\tfrac{1}{\eta+1} - \eta^2 + 2\eta + 2\right) \\
&\quad + \eta^2\left(\eta^2\tfrac{1}{\eta+1} + \eta(\tfrac{1}{\eta+1})^2 - \eta + (\tfrac{1}{\eta+1})^2 - 2\tfrac{1}{\eta+1} + 2\right) \\
&= \frac{(\eta - 2)\left(6\eta^2 + 5\eta + 2\right)}{3(\eta+1)} > 0
\end{aligned}
$$

for $\eta > 1 + \sqrt{3}$.

We then show that under $y = \tilde{y}$, $x = \tilde{x} > \frac{1}{\eta+1}$, so this case does not exist.

**Case 3.3** ($y = \tilde{y} = \frac{(\eta^3 x + \eta^2 x - \eta^2 + 2\eta + 2)}{3(\eta+1)}$, $t = 2$, $x = \tilde{x}$).

We first derive the expression for $\tilde{x}$ as a function of $\eta$ and $t$.

$$\tilde{x} = \tfrac{\eta t \tilde{y} - \eta t + t \tilde{y} + 2}{2(\eta+1)} \Rightarrow \tilde{x} = \tfrac{\eta^2 + \eta - 5}{(\eta+1)(\eta^2 - 3)}$$

However,

$$\tilde{x} - \frac{1}{\eta+1} = \frac{\eta^2 + \eta - 5}{(\eta+1)(\eta^2 - 3)} - \frac{1}{\eta+1} = \frac{\eta - 2}{(\eta+1)(\eta^2 - 3)} > 0$$

The case does not exist.

Putting Case 3.1 and 3.2 together, we have:

$$h(x, \tilde{y}, 2) \geq \min\{h(\tfrac{\eta^2 - 2\eta - 2}{\eta^2(\eta+1)}, \tilde{y}, 2), h(\tfrac{1}{\eta+1}, \tilde{y}, 2)\} \geq 0 \qquad \square$$

*Remark* B.13. Bound in Lemma B.12 is tight as a 2-competitive algorithm (i.e., *Round-Robin*) exists matching the lower bound.

*Proof for Lemma B.7.* It follows from Lemmas B.8, B.11, and B.12. $\qquad \square$

*Proof for Theorem 4.1.* It follows from Lemmas B.4 and B.7. $\qquad \square$

FIRST ORDER APPROXIMATION FOR BOUND $\lambda(\eta)$

*Remark* B.14. We have $\lambda(\eta) \geq 0.18\eta + 1.5$ for $\eta^* < \eta \leq 1 + \sqrt{3}$, so the piecewise smoothness bound in Theorem 4.1 yields $s_A(\eta) \geq 0.18\eta + 1.5$ over $(\eta^*, 1 + \sqrt{3}]$.

*Proof.*

$$\lambda(\eta) \geq 0.18\eta + 1.5$$

$$\Leftrightarrow \frac{2\eta\left(\eta^2 + \eta + \sqrt{(\eta+1)(2\eta^4 + \eta^3 + 9\eta^2 + 16\eta + 8)}\right)}{\eta^4 + 4\eta^2 + 8\eta + 4} \geq 0.18\eta + 1.5$$

$$\Leftrightarrow 2\eta\sqrt{(\eta+1)(2\eta^4 + \eta^3 + 9\eta^2 + 16\eta + 8)} \geq (0.18\eta + 1.5)(\eta^4 + 4\eta^2 + 8\eta + 4) - 2\eta(\eta^2 + \eta)$$

$$\Leftrightarrow 4\eta^2(\eta+1)(2\eta^4 + \eta^3 + 9\eta^2 + 16\eta + 8) \geq$$
$$(0.18\eta^5 + 1.5\eta^4 - 1.28\eta^3 + 5.44\eta^2 + 12.72\eta + 6.0)^2$$

$$\Leftrightarrow 8\eta^7 + 12\eta^6 + 39.82\eta^5 + 98.5\eta^4 + 97.28\eta^3 + 26.56\eta^2 - 12.72\eta - 6.0 \geq 0$$

$$\Leftrightarrow P(\eta) \geq 0$$

where:

$$P(\eta) = -0.0324\eta^{10} - 0.54\eta^9 - 1.7892\eta^8 + 9.8816\eta^7 - 10.5376\eta^6 + 13.6064\eta^5$$
$$+ 84.9696\eta^4 - 27.0336\eta^3 - 195.0784\eta^2 - 152.64\eta - 36$$

We show $P(\eta) \geq 0$ for all $\eta \in [\eta^*, 1 + \sqrt{3}]$, where $\eta^*$ is the positive root of $x^4 - 4x - 4 = 0$.

Since $P$ is a polynomial, it is continuous and differentiable over $[\eta^*, 1 + \sqrt{3}]$. On the compact interval $[\eta^*, 1 + \sqrt{3}]$, its global minimum occurs at either the boundary or at a point where $\frac{d}{d\eta} P(\eta) = 0$. We compute:

$$\frac{d}{d\eta} P(\eta) = -0.324\eta^9 - 4.86\eta^8 - 14.3136\eta^7 + 69.1712\eta^6 - 63.2256\eta^5 + 68.032\eta^4$$
$$+ 339.8784\eta^3 - 81.1008\eta^2 - 390.1568\eta - 152.64$$

Numerically solving $\frac{d}{d\eta} P(\eta) = 0$ in $[\eta^*, 1 + \sqrt{3}]$ yields one real roots:

$$\eta_1 \approx 2.475$$

We compute (to high precision):

$$P(\eta^*) \approx 25.154, \quad P(\eta_1) \approx 952.345, \quad P(1 + \sqrt{3}) \approx 305.734$$

All three values are strictly positive. The minimum of $P$ on $[\eta^*, 1 + \sqrt{3}]$ must occur at one of $\{\eta^*, \eta_1, 1 + \sqrt{3}\}$. Since $P$ is positive at each of these points, it follows that:

$$P(\eta) \geq 0 \text{ for all } \eta \in [\eta^*, 1 + \sqrt{3}] \qquad \square$$

ON SMOOTHNESS METRIC

We now demonstrate how our refined smoothness metric captures valuable information that the classical definition misses. Focus on the regime $\eta < \eta^*$, for which we proved the lower bound $s_A(\eta) \geq \eta$, with equality attained at $(x, y, e) = (\frac{1}{\eta+1}, \frac{\eta}{\eta+1}, 0)$ against the adversarial instance.

Under the classical smoothness model—where all predictions are allowed to be identical ($p_j = p$ for every $j$) and hence carry no additional information—the example tells us that any sequential ordering of the jobs might achieve the bound. Any algorithm that completes the jobs (randomly) sequentially will experience the first $\frac{1}{\eta+1}n$ ones revealed as large and the rest as small. That gives us no guidance to the optimal ordering. In contrast, our new smoothness model includes a vanishingly small tiny job, forcing an adversarial instance where finishing that job soon does matter. In this setting, the example singles out the policy that respects the job size predictions, i.e., the algorithm should process the tiny job (with the smallest job size prediction) immediately, thus recommending a scheduling policy that leverages job size predictions rather than leaving us with an arbitrary choice. Therefore, our refined metric turns a vacuous "any order works" statement into a meaningful pointer toward the *Shortest Predicted Job size First (SPJF)* scheduling.

PROOF OF THEOREM 4.2

*Proof for Theorem 4.2.* We show that *Shortest Predicted Job size First (SPJF)*, i.e., running the jobs sequentially from the smallest job size prediction to the largest, is $\eta^2$-competitive.

Fix any problem instance $I$ with prediction error $\eta$. Assume without loss of generality that $p_1^* \leq \cdots \leq p_n^*$. Define $d(i,j)$ to be the amount of processing job $J_i$ has received by the completion of job $J_j$ for any pair of jobs $(i,j)$. Then, we have:

$$A(I) = \sum_{j=1}^{n} p_j^* + \sum_{1 \leq i < j \leq n} (d(i,j) + d(j,i))$$

Observe that $d(i,j) + d(j,i) = p_i^*$ if $p_i < p_j$ and that $d(i,j) + d(j,i) \leq p_j^*$ if $p_i \geq p_j$, for any $i < j$. Notice that $p_j^* \leq \eta p_j \leq \eta p_i \leq \eta^2 p_i^*$ if $p_i \geq p_j$. We have:

$$A(I) \leq \sum_{j=1}^{n} p_j^* + \sum_{\substack{1 \leq i < j \leq n \\ p_i < p_j}} p_i^* + \sum_{\substack{1 \leq i < j \leq n \\ p_i \geq p_j}} p_j^* = \sum_{j=1}^{n} p_j^* + \sum_{1 \leq i < j \leq n} p_i^* + \sum_{\substack{1 \leq i < j \leq n \\ p_i \geq p_j}} (p_j^* - p_i^*)$$

$$\leq Opt(I) + \sum_{\substack{1 \leq i < j \leq n \\ p_i \geq p_j}} (\eta^2 p_i^* - p_i^*) \leq Opt(I) + (\eta^2 - 1) \sum_{\substack{1 \leq i < j \leq n \\ p_i \geq p_j}} p_i^* \leq \eta^2 \cdot Opt(I) \qquad \square$$

## C   PROOFS FOR SECTION 5

**Notations.** Running algorithm $A$ against problem instance $I$ produces a schedule, $\sigma_A(I)$, which specifies which machine processes which job given any time. The completion time of $J_j$ in a schedule $\sigma$ is denoted by $C_j^\sigma$. The makespan of schedule $\sigma$ is $C_{\max}^\sigma = \max_{1 \leq j \leq n} C_j^\sigma$. The cost incurred by $A$ is denoted by $A(I) = C_{\max}^{\sigma_A(I)}$. Let $\sigma^*(I)$ denote the optimal schedule, i.e., $C_{\max}^{\sigma^*(I)} = \min_{\sigma(I)} C_{\max}^{\sigma(I)}$, where $\sigma(I)$ denotes any valid schedule for instance $I$. We use $Opt(I) := C_{\max}^{\sigma^*(I)}$ to denote the cost of the optimal schedule.

*Proof for Theorem 5.1.* We partition the range for $\eta^2$ ($\eta^2 \geq 1$) into disjoint intervals and specify the adversary strategy for each interval. The partition is as follows: $\eta^2 \in \{1\} \cup (1,2] \cup (2,\frac{5}{2}] \cup (\frac{5}{2},3] \cup \ldots \cup (k - \frac{1}{k-1}, k] \cup (k, k+1 - \frac{1}{k}] \cup \ldots$, where $k \geq 2$ is an integer. We consider two generic cases.

**Case 1** ($\eta^2 \in (k - \frac{1}{k-1}, k]$)**.** We create $n = k(k-1) + 2$ jobs and $k$ machines: $p_1^* = \eta^2$ and $p_j^* = 1$, $2 \leq j \leq n-1$, with $p_j = \eta$ for $1 \leq j \leq n-1$, and $p_n^* = p_n = \epsilon$ for some small positive $0 < \epsilon < 1$. This problem instance belongs to $\mathcal{I}_s$, as $\sqrt{P} = \sqrt{\frac{p_1^*}{p_n^*}} = \sqrt{\frac{\eta^2}{\epsilon}} > \eta$. By re-indexing the jobs and machines, one can force any algorithm to allocate job $J_1$ no sooner than time $k - 1$. This is because the algorithm cannot distinguish jobs in set $S = \{J_1, ..., J_{n-1}\}$, so the adversary can set the first $n-1$ jobs (from set $S$) that have been processed by the scheduler for at least one unit of time to be size-1 jobs. Thus, the earliest time $J_1$ can start is no sooner than $\frac{k(k-1)}{k} = k - 1$. Therefore, the makespan achieved by the algorithm must be at least $k - 1 + \eta^2$. The optimal schedule is to allocate $J_1, J_n$ to $M_1$ and jobs $J_2, ..., J_{n-1}$ to the rest of the machines, each running exactly $k$ of them. The optimal makespan is at most $k + \epsilon$. We have shown the smoothness $s(A) \geq \frac{k-1+\eta^2}{k+\epsilon}$ for any $\epsilon > 0$, which gives $s(A) \geq 1 + \frac{\eta^2 - 1}{k} = 1 + \frac{\eta^2 - 1}{\lceil \eta^2 \rceil}$, where the equality is due to $\eta^2 \in (k - \frac{1}{k-1}, k]$ and $k$ is an integer. It also holds that $s(A) \geq 1 + \frac{\eta^2 - 1}{k} > 1 + \frac{k - \frac{1}{k-1} - 1}{k} = 2 - \frac{1}{k-1} = 2 - \frac{1}{\lfloor \eta^2 \rfloor}$.

**Case 2** ($\eta^2 \in (k, k+1 - \frac{1}{k}]$)**.** We create $n = k(k-1) + 2$ jobs and $k$ machines: $p_1^* = \eta^2$ and $p_j^* = \frac{\eta^2}{k}$, $2 \leq j \leq n-1$, with $p_j = \eta$ for $1 \leq j \leq n-1$, and $p_n^* = p_n = \epsilon$ for some small positive $0 < \epsilon < 1$. The following holds via similar arguments as used for case 1. The problem instance belongs to $\mathcal{I}_s$. Re-indexing the jobs and the machines, one can force any algorithm to allocate job $J_1$ no sooner than $\frac{k-1}{k}\eta^2$. Thus, the makespan achieved by the algorithm must be at least $\frac{2k-1}{k}\eta^2$. The optimal makespan is at most $\eta^2 + \epsilon$. Therefore, $s(A) \geq 2 - \frac{1}{k} = 2 - \frac{1}{\lfloor \eta^2 \rfloor}$, where the equality is due to $\eta^2 \in (k, k+1 - \frac{1}{k}]$ and $k$ is an integer. It also holds that $s(A) \geq 2 - \frac{1}{k} = 1 + \frac{k - \frac{1}{k}}{k+1} \geq 1 + \frac{\eta^2 - 1}{k+1} = 1 + \frac{\eta^2 - 1}{\lceil \eta^2 \rceil}$. $\square$

*Proof for Theorem 5.4.* Fix $\epsilon > 0$. Run *SIMPLE* (with $\epsilon/2$), treating job size predictions as job sizes with list-adjusting. We show that this *List-adjusting SIMPLE (L-SIMPLE)* satisfies the claim.

Let $\sigma_S, \sigma_{LS}, \sigma_{OPT}$ be the schedules produced by *SIMPLE* $(\epsilon/2)$, *L-SIMPLE* $(\epsilon/2)$, and the optimum. Let $C_{\max}(\sigma)$ denote the makespan of schedule $\sigma$ and $\widetilde{C}_{\max}(\sigma)$ that if the job sizes are replaced by the predictions, i.e., $C_{\max}(\sigma) = \max_{i \in [m]} \sum_{j \in N_\sigma[i]} p_j^*, \widetilde{C}_{\max}(\sigma) = \max_{i \in [m]} \sum_{j \in N_\sigma[i]} p_j$, where $[m] = \{1, ..., m\}$ and $N_\sigma[i]$ represents the jobs assigned to $M_i$ in schedule $\sigma$. We have:

$$\frac{1}{\eta} \cdot C_{\max}(\sigma_S) \leq \widetilde{C}_{\max}(\sigma_S) \leq (1 + \epsilon/2) \cdot \widetilde{C}_{\max}^*$$

$$\leq (1 + \epsilon/2) \cdot \widetilde{C}_{\max}(\sigma_{OPT}) \leq (1 + \epsilon/2) \cdot \eta \cdot C_{\max}^*$$

where $\widetilde{C}_{\max}^* = \min_\sigma \widetilde{C}_{\max}(\sigma)$ and $C_{\max}^* = C_{\max}(\sigma_{OPT})$. The first and last inequalities are due to that every job size differs by, at most, a multiplicative factor of $\eta$ in different accountings. The second inequality is by Theorem 5.3 and the third is by the optimality of $\widetilde{C}_{\max}^*$. If $\eta^2 < 2$, we have $C_{\max}(\sigma_{LS}) \leq C_{\max}(\sigma_S) < (\eta^2 + \epsilon) \cdot C_{\max}^*$, where the first inequality is due to that list-adjusting does not increase the makespan. Schedule $\sigma_{LS}$ is replicable via list scheduling with the job order of non-decreasing start time in $\sigma_{LS}$. Thus, $C_{\max}(\sigma_{LS}) \leq (2 - \frac{1}{m}) \cdot C_{\max}^*$ due to the competitive ratio of list scheduling. Finally, list-adjusting takes constant time per operation, so the run time of *L-SIMPLE* is dominated by *SIMPLE*, which is polynomial in $n$ for fixed $\epsilon$. $\qquad \square$

# D  PROOFS FOR SECTION 6

*Proof for Lemma 6.2.* The adversary's edge is that jobs $J_1, ..., J_{n-1}$ are indistinguishable from the scheduler's point of view. The adversary maintains a set of jobs $J(t)$; initially, $J(0) = \{J_1, ..., J_{n-1}\}$. The adversary monitors $x_i(t)$ per machine $M_i$ at any time $t$, where $x_i(t)$ is how long $M_i$ has been non-preemptively running a job. The adversary wakes up at time $t$ when $s_i \cdot x_i(t) = \min_{J_j \in J(t)} p_j^*$ for some machine $M_i$; it sets the index of that job to be $n - |J(t)|$ (so the job immediately finishes) and removes the job from $J(t)$. If the monitoring condition holds for more than one machine, the adversary sets the index for the job running on the smallest indexed machine first and then moves to the second smallest and so on until $s_i \cdot x_i(t) < \min_{J_j \in J(t)} p_j^*$ for all $M_i$.

This adversary strategy works since the scheduler cannot distinguish jobs in $J(t)$ at any time $t$, even if it somehow knows the job sizes but does not know which size links to which job. The scheduler can learn lower bounds of job sizes as they are processed, but every job in $J(t)$ has a size no less than these lower bounds, leaving every bijection between $J(t)$ and $L(t) := \{p_j^* | J_j \in J(t)\}$ possible. The predictions also do not reveal whether a prediction is an over- or under-estimation, even if the scheduler has $L(t)$. This is because the job size is revealed only when completed. The scheduler initially knows that some predictions are overestimations of the job sizes but does not know which until the last job with a size less than $\eta$ finishes to realize that all jobs in $J(t)$ have predictions no less than their job sizes in the future.

Property one in the claim holds immediately following the adversary strategy. We show $t_s(l) + \frac{p_j^*}{s_{m(l)}} \geq t_f(j)$ for any $j < l < n$ via proof by contradiction. Let $t_x = t_s(l) + \frac{p_j^*}{s_{m(l)}}$. If $t_x < t_f(j)$, at time $t_x$, the job $J_l$ running on machine $M_{m(l)}$ must be completed, as $s_{m(l)} \cdot x_{m(l)}(t_x) = s_{m(l)} \cdot (t_x - t_s(l)) = p_j^* \geq \min_{J_q \in J(t_x)} p_q^*$. Then, it holds $t_f(l) = t_x < t_f(j)$ contradicting $j < l$ and property one. $\qquad \square$

*Proof for Lemma 6.3.* If any job $J_j \in T_i$ is completed on a machine from $G_i$ or slower, we have $t_f(j_i^{\max}) - t_s(j) \geq t_f(j) - t_s(j) \geq \frac{2^i}{2^i} = 1$ and the claim holds. It suffices to show the claim if every job $J_j \in T_i$ is completed on a machine from $G_{i+1}$ or faster. Processing all jobs from $T_i$ on all machines in $G_{i+1}$ or faster must take more than one unit of time:

$$\max_{J_j \in T_i} \{t_f(j_i^{\max}) - t_s(j)\} \geq \frac{\sum_{J_j \in T_i} p_j^*}{\sum_{M_l \in \bigcup_{q=i+1}^{k-1} G_q} s_l}$$

$$= \frac{2^i \cdot |T_i|}{\sum_{l=i+1}^{k-1} s_l \cdot |G_l|} = \frac{2^{2k-i-1}}{\sum_{l=i+1}^{k-1} 2^l \cdot 2^{2k-2l-1}} > 1 \qquad \square$$

*Proof for Lemma 6.4.* **Case 1.** All jobs from $T_i$ are completed on machines from $G_{i+1}$ or faster. By Lemma 6.3, it takes at least one unit of time for all machines in $G_{i+1}$ or faster to process all jobs from $T_i$. Thus, there must exist one machine, named $M_r \in \bigcup_{l=i+1}^{k-1} G_l$, that processes at least one unit of time for jobs in $T_i$ by the pigeonhole principle. Thus, there exist jobs $J_j$ and $J_l$ such that $J_j, J_l \in T_i$, $m(j) = m(l) = r$ and $t_f(j) - t_s(l) \geq 1$. Running $J_l$ takes at most $\frac{1}{2}$ unit of time: $t_f(l) - t_s(l) = \frac{p_l^*}{s_r} \leq \frac{2^i}{2^{i+1}}$. We also have $t_f(j_{i-1}^{\max}) \leq t_f(l)$ by the first adversarial property. Therefore:

$$t_f(j_i^{\max}) - t_f(j_{i-1}^{\max}) \geq t_f(j) - t_f(j_{i-1}^{\max}) \geq t_f(j) - t_f(l)$$
$$\geq 1 - (t_f(l) - t_s(l)) \geq 1 - \frac{1}{2} = \frac{1}{2}$$

**Case 2.** There exists a job $J_j \in T_i$ completed on a machine from $G_i$ or slower. By the second adversarial property, we have $t_s(j) + p_{j_{i-1}^{\max}}^* / s_{m(j)} \geq t_f(j_{i-1}^{\max})$. Therefore:

$$t_f(j_i^{\max}) - t_f(j_{i-1}^{\max}) \geq t_f(j) - t_f(j_{i-1}^{\max})$$
$$= t_s(j) - t_f(j_{i-1}^{\max}) + \frac{p_j^*}{s_{m(j)}} \geq \frac{p_j^* - p_{j_{i-1}^{\max}}^*}{s_{m(j)}} \geq \frac{1}{2} \qquad \square$$

*Proof of Theorem 6.1.* If $C_{\max}^A$ is the makespan achieved by algorithm $A$, we have:

$$C_{\max}^A - t_f(j_0^{\max}) \geq t_f(j_{k-1}^{\max}) - t_f(j_0^{\max})$$
$$= (t_f(j_{k-1}^{\max}) - t_f(j_{k-2}^{\max})) + ... + (t_f(j_1^{\max}) - t_f(j_0^{\max}))$$
$$\geq \frac{k-1}{2} \quad \text{(Lemma 6.4)} \Rightarrow C_{\max}^A \geq \frac{k-1}{2} + t_f(j_0^{\max})$$

By Lemma 6.3, we have $t_f(j_0^{\max}) \geq 1$ and:

$$C_{\max}^A \geq \frac{k+1}{2} \geq \frac{[2\log\eta] + 2}{2} \geq \lceil \log\eta \rceil \qquad \square$$

