# OpenReview forum: "On Smoothness Bounds for Non-Clairvoyant Scheduling with Predictions"
_ICLR.cc/2026/Conference — ICLR 2026 Poster_

### Official Review · Reviewer_m6da · 2025-10-17

**Soundness:** 3
**Presentation:** 3
**Contribution:** 3
**Rating:** 8
**Confidence:** 4

**Summary:**

The present paper studies non-clairvoyant scheduling scheduling with predictions. In this online scheduling problem, we are given job that need to be scheduled on a single or multiple machines over time. A non-clairvoyant algorithm only learns about a job's processing time once it is completed. In particular, the paper studies this problem for minimizing the average completion time on a single machine, and minimizing the makespan on parallel identical and uniformly related machines.

In the classic online setting, these problems are well understood. The paper considers them in a learning-augmented setting, where initially a prediction on each jobs processing time is given. The quality of the prediction is measures by the maximum relative deviation factor $\eta$ of the predictions to the actual processing times. The authors propose a new idea to classify, for a fixed error value, all instances into those where the prediction error is correct, helpful, and not helpfu. Their key idea is that if their exists a prediction that achieves for the instance the fixed prediction error, then it is still useless if all predicted job lengths are the same. By that, they remove instances from the set of instance on which they want to achieve smooth guarantees, and thereby can show better bounds than previous work.

They then apply this new definition to the three aforementioned problem variants, and prove error-dependent performance guarantees depending on $\eta$.

**Strengths:**

- The paper proposes a new way to establish smoothness bounds for learning-augmented algorithms, and showcases it in non-clairvoyant scheduling, a well-studied and established class of problems in the area.
- The bounds are highly non-trivial and optimized. They improve over previous work.
- They prove lower bounds that show that previous results are best-possible
- The paper considers three different problems, which have all been studied before, and thereby nicely continues the story of learning-augmented scheduling.

I think that the paper is a nice contribution to the field of learning-augmented algorithms. It has a new conceptual idea and executes it across different settings, which showcases it applicability. It also gives a good overview over literature. I think it is a good fit for ICLR.

**Weaknesses:**

- One can argue that the results are somewhat incremental, but I think this is rather a minor weakness.
- I would prefer a slightly less technical main part. For a revised version, I would recommend to simplify or upper bound complicated expressions like in Theorem 4.1.

**Questions:**

As far as I know, the type of multiplicative prediction error has been introduced in a paper by Azar, Leonardi, and Touitou (STOC 2021). Perhaps you update these references.

---

> ### Author Response · Authors · 2025-11-18
>
> We appreciate the reviewer's positive feedback, excellent summary of the paper, and strong support for our work. Below, we provide detailed responses to your comments. We will also incorporate these insights into a revised version to improve its clarity and completeness.
>
> **Simplifying the technical main part**
>
> Thanks for your suggestions on simplifying Section 4. We will add more helper text/description to this section so it helps the reader to understand the technicalities more easily. The aim will be to provide the intuition and idea in full, and let the math be there just for precision; we will make sure that the reader can skip all the math and still get the full idea. We agree that the expression in Theorem 4.1 is complicated; it was presented as such simply because that expression was the exact tightest bound we could derive using our analysis technique. Following Theorem 4.1, we have Remark 4.3. which gives a nice bound ($0.18\eta + 1.5$) on that complicated expression. We will clarify that it is a bound for the complicated expression in Theorem 4.1 more explicitly in the revised version.
>
> **Missing references**
>
> We appreciate that the reviewer pointed out the missing reference "Flow time scheduling with uncertain processing time" by Azar, Leonardi and Touitou, STOC 2021. This is indeed relevant. We had it, but removed it to fit in the page limit for the initial submission. We will add it back in the revised version and highlight their introduction of the multiplicative prediction error.

---

> > ### Comment · Reviewer_m6da · 2025-11-20
> >
> > Dear authors,
> > thank you for your response. I appreciate that you are willing to improve the main technical part and add references. I will continue to support your paper.

---

> ### Author Response · Authors · 2025-11-25
> **Updates in the revised version**
>
> Dear Reviewer,
>
> Thank you very much for your reply.
>
> We would like to update you on the changes made in the revised version based on our previous discussion.
>
> **Changes in the revised version**
>
> To simplify the technical presentation, we have made the following revisions.
>
> We rewrite Remark 4.3 to emphasize that a simplified lower bound for $\lambda(\eta)$ is provided.
>
> >We have $\lambda(\eta) \geq 0.18\eta + 1.5$ for $\eta^\* < \eta \leq 1+\sqrt3$, so the piecewise smoothness bound yields $s_A(\eta) \geq 0.18\eta + 1.5$ over $(\eta^*, 1+\sqrt{3}]$, a simplified lower bound for $\lambda(\eta)$ in Theorem 4.1.
>
> We remove the detailed mathematical expressions when describing the adversary problem instance (Section 4) from the main text and keep them in the Appendix only. The revised text is as follows.
>
> >Consider problem instance $I$: the first $n$ jobs have non-increasing sizes with the same job size prediction $p$, and the last job has a tiny job size but the correct job size prediction. We obtain: (1) predictions reveal information, (2) the first $n$ jobs are indistinguishable, and (3) the smallest job, although visible, contributes insignificantly to the total completion time.
>
> We simplify the discussion on how the adversary constructs the adversarial instance by considering the corresponding optimization as follows.
>
> >Any algorithm $A$ will operate blindly against the first $n$ jobs, so the adversary can force the algorithm to schedule these jobs in non-increasing order of job sizes. The optimum is running the jobs in non-decreasing order of job sizes. Then, the adversary considers the following optimization.
>
> We simplify the description of the adaptive adversary strategy as follows.
>
> >Let any algorithm $A$ runs for $t^* = (\eta p) \cdot \beta^* n $ time. A job is completed if and only if it has been processed for $\eta p$ time. These jobs are called *large* jobs. At time $t^\*$, the adversary lets any job that has been processed for $p/\eta$ to finish immediately (called *intermediate* jobs) and sets the rest of the jobs to have job size $p/\eta$ (*small* jobs). We show that when the prediction error is small, this adversary pushes the algorithm to complete all the large jobs before $t^*$ and to leave no intermediate jobs. This implies the lower bound of $\eta$. When the error is large, the adversary pushes the algorithm to equally share the processor among the active jobs. This implies the lower bound of $2$. When the error is moderate, however, the algorithm may mix *SPJF* and *RR* to fight against this adversary. In this case, the lower bound implied by this adversary is $\lambda(\eta)$.
>
> We implement similar simplifications in a few other places. All changes are highlighted in blue in the revised version. While the main text now presents the ideas at a higher level, the Appendix contains the complete mathematical details.
>
> We also update the references. In particular, we now reference the works of Azar et al. after introducing the multiplicative prediction error:
>
> >This multiplicative error has also been used by Azar et al. (2021; 2022). They define the error ($\lambda$) as the product of the maximum underestimation and overestimation factors. The worst-case predictions can have these factors the same, where $\lambda=\eta^2$.
>
> Yossi Azar, Stefano Leonardi, and Noam Touitou. *Flow time scheduling with uncertain processing time*. STOC 2021.
>
> Yossi Azar, Eldad Peretz, and Noam Touitou. *Distortion-oblivious algorithms for scheduling on multiple machines*. ISAAC 2022.
>
>
> **Closing remarks**
>
> We sincerely appreciate your strong support for our work and your helpful suggestions on reducing the complexity of the main content. We are grateful for your recognition of the paper’s contributions. Your continued strong support would greatly help increase the visibility of the paper (e.g., through an oral presentation). Giving an oral presentation at a top conference has long been a personal aspiration, and we believe this work meets that standard. We would be sincerely grateful if you would consider revisiting your evaluation and, if appropriate, further raising your score.

---

### Official Review · Reviewer_88uW · 2025-10-25

**Soundness:** 3
**Presentation:** 3
**Contribution:** 2
**Rating:** 6
**Confidence:** 4

**Summary:**

This paper studies the non-clairvoyant scheduling problem in an online setting under the algorithms with predictions framework.  In addition to the standard notions of consistency and robustness, the authors revisit the notion of smoothness, which captures the performance of a learning-augmented algorithm when predictions are in the middle ground — not perfectly accurate but also not completely adversarial (i.e., there is a bounded prediction error).  This problem has been studied before, but the authors propose a different (superior) metric to measure prediction error, and upper / lower bounds on algorithm performance in three scenarios using this error metric.

**Strengths:**

The proposed method to measure prediction error (multiplicatively rather than as the $L_1$ difference) is natural and makes sense for the problem.

Smoothness bounds are known to be somewhat difficult to obtain in the literature on learning-augmented algorithms (in the sense that many papers omit it completely and focus on the sometimes easier to analyze consistency and robustness metrics).  This paper provides a useful definition of smoothness (as a partitioning of the universe of inputs) that could help to model some notion of smoothness in other problems.

The paper studies several settings: single machine scheduling, parallel machines scheduling, and uniform machines scheduling, and provides or completes a set of upper and lower bounds for each setting.

**Weaknesses:**

Figure 2 is quite hard to read in its current form due to small text.

The upper bounds and lower bounds are not all tight in the sense that the proposed algorithms do not attain the theoretically best possible smoothness — this is a relatively mild weakness since attaining optimality in this sense seems quite challenging in the literature on related problems.

Typically at an ICLR-type of conference I would expect to see some small numerical experiments in addition to the theory.  This might also allow for some comparison between the proposed algorithms and the algorithms that rely on a different prediction error metric in the literature.

It may be worth clarifying somewhere early on that $\eta \geq 1$ in your multiplicative model, because from the abstract a reader may get the impression that your upper bound is somehow below your lower bound (i.e., if $\eta$ was allowed to be $< 1$, $\eta^2 < \eta$).

**Questions:**

The refined notion of smoothness requires this definition of instances on which \eta is small enough to provide information (reasonable predictions, Definition 3.1).  Can you speak on whether this notion of smoothness can extend to other online problems?  I know there are some problems such as one-way trading that exhibit a brittleness [1] — even a prediction with a very small prediction error can be completely uninformative in some cases.  Would this pose a challenge for the proposed partitioning?

Am I understanding correctly that the constant vector $\mathbf{p}$ mentioned around line 193 is a vector where any arbitrary element is the same value (e.g., p[0] = p[1] = p[2] = ….)?

Can you speak to the utility of smoothness bounds in practice?  They initially strike me as a primarily theoretical exercise since a prediction model’s error is not generally known apriori.  This question doesn’t detract from the paper’s contribution, but I wonder if there is some secondary benefit from considering smoothness (e.g., even if the bounds themselves are not informative in practice, perhaps the algorithms designed with smoothness in mind are better at dealing with prediction errors in practice?)

[1] On Tradeoffs in Learning-Augmented Algorithms, Ziyad Benomar, Vianney Perchet, arXiv:2501.12770

---

> ### Author Response · Authors · 2025-11-18
>
> We appreciate the reviewer's constructive comments, insightful questions, and positive feedback regarding our work. We are grateful for the opportunity to discuss the points raised.
>
> **Extending the smoothness definition to other online problems**
>
> Thanks for highlighting the extension of the refined smoothness definition to other online problems. Indeed, it is applicable; one of the visions of this work is to provide refined metrics for consistency, smoothness, and robustness that are potentially applicable to other problems. Note that different online problems and prediction settings naturally require different interpretations of predicate $R(I)$ (introduced in line 193 used to define these metrics), which captures the notion of whether the prediction might potentially not reveal any information. Take the one-way trading problem as an example. Following the notations used in [1], one candidate definition of $R(I)$ is $ | \hat{p} - p_{\sigma}^* | \geq \max \(p_{\sigma}^* - 1, M - p_{\sigma}^* \) $, where $\sigma$ is the input sequence of exchange rates, $ p_{\sigma}^* $ is the maximum rate in $\sigma$, $\hat{p}$ is the maximum-rate prediction, $| \hat{p} - p_{\sigma}^* |$ is the prediction error ($L_1$ error as used in [1]), and $M$ is an upper bound on the rates that is known in advance ($p_i \in [1, M]$ where $p_i$ denotes the $i$-th rate in the sequence). The intuition is the following: if $R(I)$ holds for some problem instance $I$, the actual $ p_{\sigma}^* $ can take any value in $[1, M]$, given the prediction $\hat{p}$, which effectively gives no additional information, as we know $p_{\sigma}^* \in [1, M]$ upfront. This predicate $R(I)$ allows one to partition the problem instances, thus leading to concrete definitions of consistency, smoothness, and robustness following the setup in our work.
>
> The refined smoothness tries to capture the performance of a learning-augmented algorithm under which the predictions reveal additional information; the partitioning of problem instances is based on whether the predictions add additional information. Brittleness, on the other hand, captures the existence of a problem instance with predictions of small error that leads a given algorithm to its robustness guarantee. These two notions capture different perspectives, but studying their relationship seems an interesting direction and is a very open domain.
>
> This discussion on the extension of our work to other online problems is valuable because it shows the broader impact of this work. We will add this discussion in the revised version to clarify the potential applicability of the proposed conceptual framework to other online problems with predictions.
>
> [1] Overcoming brittleness in pareto-optimal learning-augmented algorithms, Alex Elenter, Spyros Angelopoulos, Christoph Dürr, and Yanni Lefki, NIPS '24.
>
> **Constant vector**
>
> Yes, the constant vector means a vector where any arbitrary element is the same value (e.g., $p[0] = p[1] = p[2] = ...$). It is defined in line 194. We will make this definition more explicit in the revised version.
>
> **Utility of smoothness bounds in practice**
>
> There are two key advantages. The first one is for upper bounds. Though smoothness bound in general is a challenging bound to establish in theory, it is still a better bound (than consistency and robustness) to optimize in practice. This is because, in practice, one will likely be dealing with predictions with some errors but not adversarial, so neither consistency nor robustness describes the performance best matching the practice. While it is hard to know the prediction error apriori, it is not too hard to estimate the range of errors in practice (e.g., through historical backtesting), so practitioners might want to optimize an algorithm's smoothness over the estimated range of errors. The second advantage is for lower bounds. Roughly speaking, lower bounds indicate the informational advantages carried by the predictions, so they inform the marginal gain of improving the accuracy of predictive models from a business point of view. To see this, suppose we establish a smoothness lower bound of $\Omega(\eta)$ for some problem; that means improving the prediction error by a factor of a half may potentially give a doubled improvement on the end goal of the business; this may inform the business decision of allocating more resources to building accurate predictive models.
>
> Thanks to the reviewer for initiating the discussion on the practicality of considering smoothness. We find the discussion valuable and worth adding to the revised version of the paper, which should further motivate the study of smoothness bounds.
>
> **Clarifications for presentation**
>
> Regarding the weaknesses pointed out in 1 and 4, we will increase the font in Figure 2 and clarify that $\eta \geq 1$ (in the multiplicative model) earlier in the revised version for clarification.

---

> > ### Comment · Reviewer_88uW · 2025-11-20
> > **Reviewer Response**
> >
> > Dear authors,
> >
> > Thank you for your points of clarification and for engaging in the discussion.  I think these points will be a strong addition to the paper and emphasize the paper's contributions to the broader field of learning-augmented algorithms.
> >
> > I maintain my positive score.

---

> ### Author Response · Authors · 2025-11-25
> **Updates in the revised version**
>
> Dear Reviewer,
>
> Thank you very much for your reply.
>
> We would like to update you on the changes made in the revised version based on our previous discussion.
>
> **Changes in the revised version**
>
> To explain how our proposed smoothness definition can apply to other online problems (e.g., one-way trading), we add a new Discussion section before the Conclusion. The added content is the following.
>
> >We highlight that our proposed notion of smoothness applies to other online problems. Different online problems and prediction settings require different interpretations of predicate $R(I)$ in the definition. Take the one-way trading problem as an example (Elenter et al., 2024; Benomar and Perchet, 2025). Following the notations in Elenter et al. (2024), one candidate definition of $R(I)$ is $ | \hat{p} - p_{\sigma}^* | \geq \max (p_{\sigma}^* - 1, M - p_{\sigma}^* ) $, where $\sigma$ is the input sequence of exchange rates, $ p_{\sigma}^* $ the maximum rate in $\sigma$, $\hat{p}$ the maximum-rate prediction, $ | \hat{p} - p_{\sigma}^* | $ the prediction error, and $M$ an upper bound on the rates known upfront. The intuition is that if $R(I)$ holds for some problem instance $I$, $ p_{\sigma}^* $ can take any value in $[1, M]$, given $\hat{p}$, which effectively gives no additional information, as we know $ p_{\sigma}^* \in [1, M] $ upfront. This predicate $R(I)$ allows one to partition the problem instances, thus leading to concrete definitions of consistency, smoothness, and robustness following the setup in our work.
>
> To clarify constant vectors, we make the following revision in Section 3.
>
> >The predicate checks if any constant vector of job size predictions exists with error $\eta(I)$, where a constant vector means any arbitrary element is the same value.
>
> To better motivate the study of smoothness, we add the following discussion at the end of Section 3 after we introduce smoothness.
>
> >The authors believe smoothness is a better metric (than consistency and robustness) to consider. This is because, in practice, one will likely be dealing with predictions having some errors but not adversarial, so neither consistency nor robustness best describes the performance that matches the practice. While it is hard to know the prediction error apriori, it is not so to estimate the range of errors (e.g., through historical backtesting), so practitioners can optimize smoothness over the estimated range of errors. Another advantage is on lower bounds. Roughly speaking, lower bounds indicate the informational advantages carried by the predictions; they inform the marginal value of improving the accuracy of predictive models from a business's perspective. To see this, suppose we establish a smoothness lower bound of $\Omega(\eta)$ for some problem; that means improving the prediction error by a factor of a half may potentially give a doubled improvement on the end goal of the business; this may inform the business decision of allocating more resources to building accurate predictive models.
>
> We have updated Figure 2 with an increased font size and stated in the Abstract that $\eta \geq 1$ for clarification.
>
> Alex Elenter, Spyros Angelopoulos, Christoph Dürr, and Yanni Lefki. *Overcoming brittleness in pareto-optimal learning-augmented algorithms*. NIPS 2024.
>
> Ziyad Benomar and Vianney Perchet. *On tradeoffs in learning-augmented algorithms*. PMLR 2025.
>
>
> **Closing remarks**
>
> We are grateful for your support of our work, for your suggestions that improve the clarity of the presentation, and for your insightful questions that allow us to better communicate the motivation for studying smoothness and the broader impact of our results. We believe the paper makes a non-trivial contribution by formalizing smoothness and deriving meaningful bounds for concrete scheduling problems. Though some gaps between the lower and upper bounds remain open, our efforts represent a non-trivial attempt to narrow these gaps and show the potential of learning-augmented algorithms. Meanwhile, we provide a conceptual framework applicable beyond scheduling and may be useful for other online problems.
>
> We hope that our revisions and responses address your concerns and questions and further highlight the motivation and broader impact of our work. We would be truly grateful if you would consider revisiting your evaluation and, if appropriate, raising your score.

---

### Official Review · Reviewer_3VjR · 2025-10-28

**Soundness:** 3
**Presentation:** 2
**Contribution:** 4
**Rating:** 8
**Confidence:** 5

**Summary:**

The paper considers 3 scheduling problems. Jobs are given, but their length is revealed only at their completion. This is called the non-clairvoyant model. The 3 scheduling problems are

1. Single machine. The goal is to minimize the sum of completion times.
2. Identical parallel machines. The goal is to minimize the maximum completion time.
3. Related parallel machines. Same objective. But machines have different speeds. Execution time is job length divided by machine speed.

This model is augmented with predictions on the job length. The error between the prediction and the actual job length is measured by $\eta$, the maximum over all jobs j, of the maximum over the ratios $p_j/p^\star_j$ and $p^\star_j/p_j$, where $p_j$ is the predicted and $p^\star_j$ the actual length. Note that if the error is 1, the optimal solution could be computed, at least in exponential time for the parallel machine cases.

The performance of a schedule is measured by the competitive ratio, which compares its objective value with the optimal solution, one could compute if the actual processing times were known.

This paper is part of an active research area on learning augmented algorithms. There an important question is how to design algorithms with a competitive ratio degrading smoothly with the error.

The paper distinguishes 3 kind of instances, where an instance is the pair $\langle p, p^*\rangle$.

- $I_c$ is the set of instance with no error, $\eta(I)=1$.
- $I_r$ is the set of instance with error, for which a constant prediction $p'$ exists, that is $p'_j$ is independent of $j$, and with the same error, that is $\eta(p,p^\star)=\eta(p',p^\star)$.
- $I_s$ are all other instances.

The competitive ratio within each of these instance classes is called

- consistency
- robustness
- smoothness

The smoothness ratio is a function of eta, and the paper provides bounds for it. For the single machine problem: when eta is between 1 and 1.835, the smoothness is at least eta. When it is between 1.835 and $1+\sqrt3$, it is at least some specific function $\lambda(\eta)$. And above $1+\sqrt3$ it is at least 2. This lower bound is continuous. It is also shown that in all cases the smoothness of the _Shortest Processing Job First_ algorithm is at most $\eta^2$. Its analysis relies on previous work. There is a long discussion on how the worst case instances look like.

There are also lower and upper bounds given for the identical and the uniformly identical parallel machine scheduling problem. It improves over a work from 2023. For the identical parallel machine problem, there is an algorithm called SIMPLE from the 80's, which is 1+epsilon consistent and eta^2 robust, but not smooth. A modification is proposed where once a machine finishes all jobs assigned to it, steels pending jobs from other machines.

For the uniform parallel machine problem, an upper bound was know in 2024, and the paper provides an essentially matching lower bound.

**Strengths:**

Overall I think that this is a great paper. It is novel to distinguish instances where the prediction can actually tell something about the instance, in addition to a distinction by the prediction error. Then the results are good, sometimes tight. The results are not technically involved, but it does not always need to be the case in a good paper. The studied problems are among the most important online problems.  It leaves open problems, and hence will drive the community in an interesting direction.

Prediction error was mostly studied with respect to absolute difference. Considering the ratios for the error is in my opinion the better approach.

**Weaknesses:**

I had to read the definition of R(I) several times before understanding its deep meaning.

**Questions:**

Page 4 line 198, I found definition of $I_c$ confusing and propose $I_c=\\{I|\eta(T)=1\\}$.

I found the definition of $I_c, I_r, I_s$ not well explained. I would say that R(I) is the set of instances, for which there is a constant vector $q$ such that $\eta(p,q)=\eta(p,p^\star)$. In other words for a fixed prediction error, the set of possible actual job length vectors $p^\star$ include a constant one. Which means that the prediction error does not allow to partially order jobs according to their length, or even partition them according to the approximate length. These informations would be necessary to design a good schedule. Also I need some discussion on how this smoothness is related to the usual studied notion of the competitive ratio degrading smoothly with the prediction error.

Theorem 4.1 is hard to parse, because $\eta^*$ is defined at the end of the sentence.

I cannot parse Figure 3 on a black and white printout.

Page 8 line 386. For large error eta, the lower bound is 2, but the upper bound is 2-1/m. Hence something is not quite right.

---

> ### Author Response · Authors · 2025-11-18
>
> We appreciate the reviewer's positive feedback, excellent summary of the paper, and strong support for our work. Below, we provide detailed responses to your comments. We will also incorporate these insights into a revised version to improve its clarity.
>
> ### **Definition of $R(I)$**
>
> We appreciate the reviewer's suggestion on making the definition of $R(I)$ easier to understand. We agree that it is indeed a non-trivial concept. To improve the presentation, we will add more description before presenting the formal definition of $R(I)$, with the intention that the expression becomes a math translation of the notion (described in text). We would also like to use the suggested interpretation to achieve this. Here is our proposed description before the introduction of $R(I)$ in the revised version.
>
> For a fixed prediction error $\eta$, if one can create the job size predictions with 1) error equal to $\eta$, and 2) all predictions have the same value, the predictions will not allow one to partially order jobs, or even partition them according to the predictions. That is, the predictions are as if they are not present. In other words, for the given predictions, the set of possible actual job size vectors includes a constant one, meaning that all permutations of the jobs according to their job sizes remain possible. In such a scenario, the predictions reveal no additional information.
>
> ### **Relationship between the proposed smoothness and the existing notion**
>
> Thanks to the reviewer for pointing out this connection. The usual notion of the competitive ratio degrading smoothly with the prediction error, which exists before this work, requires an error-dependent competitive ratio, i.e., a bound written as a function of the error, to be derived for an algorithm. Our smoothness explicitly defines such a function. One can then say that an algorithm is smooth if its smoothness is in a low-order form of $\eta$, e.g., $\log \eta$ or a polynomial in $\eta$. In addition, the lower bounds indicate the limit of how such smoothness can go. If a $\Omega(\log \eta)$ bound exists, for example, it is impossible to attain a better asymptotic functional form for the competitive ratio as a function of the error, even if the predictions are revealing additional information. Such an explicitly defined smoothness allows one to study the limit of the informational advantages carried by the predictions for learning-augmented algorithms.
>
> We find that the discussion is valuable as it connects the proposed smoothness definition and the existing notion of competitive ratio degrading smoothly with the prediction error. We will add this discussion in the revised version to clarify the connection between this work and the prior development.
>
>
> ### **Clarification for presentation**
>
> **Definition of $\mathcal{I}_c$.** The current $\mathcal{I}_c$ defined in such a form is with the intention that the proposed setting for consistency, smoothness, and robustness can be used with other forms of the prediction error. In immediate line 199, we define $\mathcal{I}_c = \\{ I |~\eta(I) = 1 \\}$ with respect to the multiplicative error.
>
> **Clarifying Theorem 4.1.** Thanks for pointing this out. We will re-arrange the text to make $\eta^*$ appear at the beginning of the sentence.
>
> **Clarifying Figure 3.** Thanks for pointing this out. We will edit Figures 3 and 4 so that the plotted functions can be identified on a black and white printout.
>
> ### **Clarification on the lower bound $2$ and upper bound $2-\frac{1}{m}$**
>
> Thanks for the opportunity to clarify this point. These bounds are consistent. The lower bound holds under any arbitrary number of machines, while the upper bound holds for a fixed number of machines $m$. The lower bound, while approaching 2, is always below $2$, as the error $\eta$ increases large but stays finite; the bound never hits 2. When the error is large, the lower bound, while approaching 2, is obtained by an instance with a large number of machines (so large $m$); this can be quickly verified on lines 1543 and 1557 in the proof of Theorem 5.1, where the adversarial instance for a large error requires a large number of machines. In that case, the upper bound is also approaching $2$ but is above the lower bound.

---

> ### Author Response · Authors · 2025-11-25
> **Updates in the revised version**
>
> Dear Reviewer,
>
> We would like to update you on the changes made in the revised version in response to your comments and suggestions.
>
> **Changes in the revised version**
>
> To better communicate the intuition behind the predicate $R(I)$, we have added the following text before its definition in Section 3.
>
> >We will introduce a predicate $R(I)$ to model the worst-case predictions. It takes a problem instance as input and outputs true or false, indicating whether the predictions reveal no additional information. Before presenting its definition, we give its high-level intuition. For a fixed prediction error $\eta$, if one can create the job size predictions with 1) error equal to $\eta$, and 2) all predictions have the same value, the predictions will not allow one to partially order jobs, or even partition them according to the predictions. That is, the predictions are as if they are not present. In other words, for the given predictions, the set of possible actual job size vectors includes a constant one, meaning that all permutations of the jobs according to their job sizes remain possible. In such a scenario, the predictions reveal no additional information.
>
> Following your suggestion, we added a paragraph in Section 3 to discuss how the smoothness metric is related to the usual studied notion of the competitive ratio degrading smoothly with the prediction error.
>
> >Here, we discuss the connection between our proposed smoothness and the notion of the competitive ratio degrading smoothly with the prediction error, which exists before this work. The existing notion requires an error-dependent competitive ratio, i.e., a bound written as a function of the error, to be derived for an algorithm. Our smoothness explicitly defines such a function. One can then say that an algorithm is smooth if its smoothness is in a low-order form of $\eta$, e.g., $\log \eta$ or a polynomial in $\eta$. In addition, the lower bounds indicate the limit of how such smoothness can go. If a $\Omega(\log \eta)$ bound exists, for example, it is impossible to attain a better asymptotic functional form for the competitive ratio as a function of the error, even if the predictions are revealing additional information. Such explicit smoothness allows one to study the limit and potential of the informational advantages carried by the predictions for learning-augmented algorithms.
>
> We have rewritten Theorem 4.1 to improve readability. The revised version states the following.
>
> >If $A$ is an algorithm for a single machine to minimize total completion time and $\eta^\*$ is the positive root of $x^4 - 4x - 4 =0$ ($\eta^* \approx 1.835$), $s_A(\eta) \geq \eta$ for $\eta \leq \eta^\*$, $s_A(\eta) \geq \lambda(\eta)$ for $\eta^* < \eta \leq 1 + \sqrt{3}$, and $s_A(\eta) \geq 2$ for $\eta > 1 + \sqrt{3}$, where $\lambda(\eta) = \frac{2\eta \left(\eta^2 + \eta + \sqrt{(\eta+1)(2 \eta^4 + \eta^3 + 9\eta^2 + 16\eta +8)} \right)}{\eta^4 + 4\eta^2 + 8\eta + 4}$.
>
> Figures 3 and 4 have been revised so that each function is plotted with distinct line styles and markers to ensure clarity in black and white printouts.
>
>
> **Closing remarks**
>
> We sincerely appreciate your strong support for our work and your helpful suggestions on improving the clarity of the paper. We believe the paper discusses some important aspects of learning-augmented algorithms by (1) clarifying fundamental metrics via a partitioning of problem instances according to the information carried by predictions, and (2) demonstrating the potential of learning-augmented algorithms via, for example, showing that one can simultaneously achieve asymptotically optimal consistency, smoothness, and robustness in one algorithm.
>
> We believe these contributions are of interest to a broad audience. Your continued strong support would greatly help increase the visibility of the paper (e.g., through an oral presentation). Giving an oral presentation at a top conference has long been a personal aspiration, and we feel that this work meets that standard. We would be sincerely grateful if you would consider revisiting your evaluation and, if appropriate, further raising your score.

---

### Official Review · Reviewer_PzFT · 2025-10-30

**Soundness:** 3
**Presentation:** 2
**Contribution:** 2
**Rating:** 2
**Confidence:** 3

**Summary:**

The paper considers non-clairvoyant scheduling with job size predictions for two different objective functions, total completion time minimization on a single machine and makespan minimization on identical and uniformly-related machines. In non-clairvoyant scheduling, the actual job sizes are unknown and only revealed once a job is completed. In the setting with predictions, we have a-priori access to predictions on the job sizes. Such learning-augmented algorithm are studied with respect to their consistency (the competitive ratio for correct predictions), their robustness (the competitive ratio for arbitrarily bad predictions) and their smoothness (an guarantee depending on a prediction error).

In this paper, the authors propose an alternative definition of smoothness, which only requires error-dependent guarantees for predictions that "reveal additional information". The author study this notion for the error $\eta$, which is the maximum ratio over all jobs between a predicted job size and an actual job size. For minimising the total completion time on a single machine, the authors prove an upper bound of $\eta^2$ and a lower bound of $\eta$. For minimising the makespan on identical machines, the authors show a lower bound of $2-\mathcal{O}(\eta^{-2})$ and an upper bound of $min(\eta^2,2)$. For minimising the makespan on uniformly related machines, they show a lower bound of $\lceil\log \eta \rceil$

**Strengths:**

* The paper gives more fine-grained error-dependent lower bounds for multiple non-clairvoyant scheduling problems. These lower bounds are non-trivial. In my opinion, error-dependent lower bounds are not sufficiently studied in the area of learning-augmented algorithms. Hence, I think that the given lower bounds are a significant contribution.

**Weaknesses:**

* I don't completely get the idea behind the partition of instances in the proposed smoothness definition. Smoothness bounds according to the new definition exclude predictions that "reveal no additional information". However, if the predictions reveal no additional information, then lower bounds for the original problem without predictions should always apply. Hence, even using the original definitions, such predictions should always lead to the algorithms guarantee ending up in the robustness case, the same as in the new definition.
* The paper is missing a discussion of related work on flow time scheduling with predictions, e.g., "Distortion-Oblivious Algorithms for Scheduling on Multiple Machines" and "Flow time scheduling with uncertain processing time" by Azar, Leonardi and Touitou, and "Distortion-Oblivious Algorithms for Scheduling on Multiple Machines" by Azar, Peretz and Touitou. These works consider a more general problem, which contains total completion time minimization. Their distortion error is related to the multiplicative error that is considered in this paper. In particular, the upper bounds presented in these works should imply (weaker) upper bounds for the error measure used in this paper.
* The algorithmic results of Theorem 4.2 and Theorem 5.4 are not actually using the proposed new definition of smoothness. Instead, Theorem 4.2 is just an error-dependent bound and Theorem 5.4 is the minimum between an error-dependent bound and a robustness bound. Hence, both theorems just follow the original notions of consistency, robustness and smoothness. The proposed alternative notion of smoothness seems to only be used to give more fine-grained lower bounds. While these lower bounds are nice results, I think that algorithmic results are necessary to make a good case for establishing the proposed alternative notion of smoothness.
* The algorithmic results given in this paper are not very technically involved. I think the paper would benefit from more clearly stating that the main results are more fine-grained error-dependent lower bounds.

**Questions:**

* Could you elaborate on the advantages of your smoothness definition, taking the first weakness above into account?

---

> ### Author Response · Authors · 2025-11-18
>
> We appreciate the reviewer's constructive comments and positive remarks on the paper's contributions in the given lower bounds. We understand that the reviewer's primary concern is the necessity and advantage of our refined smoothness definition. We want to address this concern by highlighting the value-add of our refined smoothness. The following discussion aims to address the concerns raised in Weaknesses 1, 3, 4, and the question in the review.
>
>
> ### **Necessity and advantage of our refined definition of consistency, smoothness, and robustness**
>
> **Necessity and motivation.** We agree that, without the new definition, lower bounds for the original problem without predictions always apply for the case when predictions do not reveal information, which is captured by the existing robustness and the robustness metric in our setup. However, our refined definition of consistency, smoothness, and robustness is necessary because it distinguishes these metrics through the partitioning problem instances. To our knowledge, such an explicit informational setup is the first to define these metrics, so it at least introduces a new way of looking at them and meanwhile gives a clear cut between smoothness (competitive ratio as a function of the prediction error) and robustness (competitive ratio as some expression independent of the prediction error). In prior works, when smoothness is simply defined to be the competitive ratio written as a function of the prediction error, robustness can always be, by definition, interpreted as smoothness, thus blurring these two metrics. Our refined definition is an attempt to make a clear cut between these metrics so the analysis can give a clearer insight into the algorithm's performance when not facing the worst-case predictions.
>
> **Advantage of the proposed definition of smoothness.** The proposed definition formulates the notion of "the limit of the information advantages carried by the predictions (even) if they reveal additional information." The advantage naturally is in the lower bounds of smoothness. We agree that Theorem 4.2 and Theorem 5.4 (both upper bounds results) hold for both our proposed definition of smoothness and the existing notion, where one includes the problem instances with the worst-case predictions. The lower bounds results for the three problems (Theorems 4.1, 5.1, and 6.1), however, are all established using the new definition that requires the adversarial instances to have predictions that must reveal additional information. This allows one to interpret, for example, an $\eta$ smoothness lower bound as: "no algorithm can achieve better than $\eta$-smoothness even if the predictions are revealing additional information." Such an interpretation, on the one hand, reveals the fundamental limitation of predictions in these problems and, on the other hand, indicates how far we can still push learning-augmented algorithms for a better exploitation of predictions.
>
> In summary, while we believe that there could be a better approach to define these three metrics to be found in the future, our attempt is to 1) give a clear cut between consistency, smoothness, and robustness, and 2) implement the interpretation of a smoothness lower bound to be the best possible error-depedent competitive ratio even if the predictions are revealing additional information.
>
> A follow-up discussion is that our lower bounds also apply for a smoothness definition that includes the worst-case predictions, because the problem instances considered in our setup are a subset of the case that includes the worst-case predictions. Therefore, all the smoothness results still apply. This consequence is implicit. We will add a discussion in the revised version so our contributions on the smoothness bounds are more clearly communicated. The discussion will highlight 1) the established bounds hold for both the new definition and the existing notion, and 2) we establish more fine-grained error-dependent lower bounds (i.e., the limit of the information advantages carried by the predictions even if they reveal additional information).
>
>
> ### **Missing references**
>
> We appreciate that the reviewer pointed out the missing references "Flow time scheduling with uncertain processing time" by Azar, Leonardi and Touitou, and "Distortion-Oblivious Algorithms for Scheduling on Multiple Machines" by Azar, Peretz and Touitou. These are indeed relevant. We had them, but removed them to fit in the page limit for the initial submission. We will add both back in the revised version and highlight the relationship between these results and ours.

---

> > ### Comment · Reviewer_PzFT · 2025-11-20
> >
> > Dear authors,
> >
> > Thank you for your very helpful reply!
> >
> > Given your response, I realise that my first weakness completely misses the point of your definition. I pointed out that lower bounds for algorithms without predictions apply when the predictions do not reveal any information. However, this actually motivates removing such instance from the smoothness definition to, in your words, give a clear cut between consistency, robustness and smoothness. Thank you for pointing this out.
> >
> > Regarding my third weakness, I agree with your point that your lower bounds are stronger (or at least not weaker) because they hold “even if the predictions are revealing additional information”. I still think that algorithmic results are necessary to fully establish a new definition. Removing instances with predictions that do not reveal any information from the smoothness definition could in principle allow stronger smoothness upper bounds compared to the original definition. Such upper bounds would fully separate both definitions. As far as I understand, your results do not achieve this separation. However, I do agree with your point that the definitions of the three measures can be improved and that your work is a step in that direction.
> >
> > I will raise my score to a 6.

---

> > > ### Author Response · Authors · 2025-11-25
> > > **Updates in the revised version and continued discussion**
> > >
> > > Dear Reviewer,
> > >
> > > Thank you very much for your reply and for raising your score.
> > >
> > > We would like to update you on the changes made in the revised version based on our previous discussion, and to continue the conversation regarding the definition of smoothness (regarding *Weakness* 3).
> > >
> > > **Changes in the revised version**
> > >
> > > To better communicate our contributions on the smoothness definition and the bounds we derive, we have added a new Discussion section before the Conclusion. The content is the following.
> > >
> > > >Our definition of smoothness excludes the problem instances with predictions that reveal no additional information. An alternative way to define smoothness would be to include these instances, i.e, $s_A'(\eta) = \text{sup}_{I \in \\{I| I \notin \mathcal{I}_c\\}, \eta(I) \leq \eta} {\tfrac{A(I)}{Opt(I)}}$. While this also separates smoothness and robustness, we choose our definition because 1) it fully separates the problem instances considered by the two metrics, 2) the smoothness lower bound has the interpretation that "no algorithm can achieve better than $s_A(\eta)$ even if the predictions are revealing additional information", and 3) the smoothness bounds are stronger. All the derived bounds (Theorems 4.1, 4.2, 5.1, 5.4, and 6.1) still hold for $s_A'(\eta)$, because the problem instances considered in our definition are a subset of those in $s_A'(\eta)$. It remains open if the proposed smoothness definition makes a difference in the upper bounds, i.e., if there exists an algorithm $A$ to some problem where $s_A(\eta) < s_A'(\eta)$.
> > >
> > > We also update the references. In particular, we now reference the works of Azar et al. after introducing the multiplicative prediction error.
> > >
> > > >This multiplicative error has also been used by Azar et al. (2021; 2022). They define the error ($\lambda$) as the product of the maximum underestimation and overestimation factors. The worst-case predictions can have these factors the same, where $\lambda=\eta^2$.
> > >
> > > At the beginning of Section 4, we state how their results imply an upper bound for completion time scheduling.
> > >
> > > >An upper bound that can be immediately obtained from Azar et al. (2022) is $O(\eta^2 \log \eta)$. We show the following lower and (improved) upper bounds on smoothness.
> > >
> > > Yossi Azar, Stefano Leonardi, and Noam Touitou. *Flow time scheduling with uncertain processing time*. STOC 2021.
> > >
> > > Yossi Azar, Eldad Peretz, and Noam Touitou. *Distortion-oblivious algorithms for scheduling on multiple machines*. ISAAC 2022.
> > >
> > >
> > > **Continued discussion on the smoothness definition**
> > >
> > > We would like to clarify that two (separate) aspects are involved in the definition:
> > > 1. separating smoothness and robustness; and
> > > 2. deciding whether smoothness should include instances with predictions having no additional information (the worst-case predictions).
> > >
> > > The alternative smoothness definition you have in mind (which we denote by $s_A'(\eta)$ in the Discussion section) also addresses point (1), because both the alternative definition and ours define smoothness as a function of $\eta$, but robustness does not. Thus, both definitions give a clear cut between smoothness and robustness. We choose our definition because it (i) separates the problem instances underlying smoothness and robustness, (ii) has a more meaningful interpretation, and (iii) is stronger. We acknowledge that these criteria are somewhat discretionary, but the primary goal is to refine the three metrics and give a clear cut between them, so point (2), whether to include worst-case predictions, is relatively minor. Importantly, the bounds derived under our definition hold under the alternative definition, but the reverse does not hold in general. This leads to the following argument.
> > >
> > > In addition to our first contribution in refining the notions of consistency, robustness, and smoothness, we derive non-trivial smoothness bounds for the three concrete scheduling problems. As the reviewer has commented, such error-dependent lower bounds are not sufficiently studied in the area of learning-augmented algorithms, and we believe they form a good part of our contribution. We would like to reiterate that these bounds apply under both our definition and the alternative, and, therefore, the alternative smoothness definition should not weaken the contributions of the derived bounds.

---

> ### Author Response · Authors · 2025-11-25
>
> **Closing remarks**
>
> We are grateful for your support of our work. We understand your point regarding upper bounds that separate the two possible smoothness definitions. While finding such an instance is an interesting open direction, we view it as a secondary issue in our work and believe our current results already make a good contribution by clarifying consistency, robustness, and smoothness, and deriving bounds showing the limits and potential of learning-augmented algorithms. In particular, for uniformly-related machines scheduling, we show that it is possible to obtain asymptotically optimal consistency, robustness, and smoothness all at once; this shows the potential of well-designed learning-augmented algorithms.
>
> We believe that our works add useful insights to the community of learning-augmented algorithms, and we would greatly appreciate your strong support in increasing the visibility of this work. We hope our responses and revisions have addressed your concerns. We would be truly grateful if you would consider revisiting your evaluation and, if appropriate, further raising your score.

---

### Author Response · Authors · 2025-11-30
**Summary Response to All Reviewers and Area Chair**

We sincerely thank all reviewers for their evaluation, feedback, and supporting recognition of our contributions. Below, we summarize our work, the reviewer's comments, the discussion, and the broader impact of our work. Our work studies learning-augmented algorithms, a type of algorithms that make decisions based on (possibly erroneous) predictions of some unknowns. It introduces refined definitions of consistency, smoothness, and robustness. Based on these definitions, it establishes smoothness bounds for three fundamental scheduling problems.

----

**Main positive aspects highlighted by reviewers**

- Novel definitions of consistency, smoothness, and robustness based on whether predictions provide additional information (Reviewers 3VjR, 88uW, m6da).
- Non-trivial, optimized, and in some cases tight smoothness bounds for the considered problems (All reviewers).
- Use of multiplicative prediction error, a more appropriate metric for the considered problems (Reviewers 3VjR, 88uW).


**Main concerns raised**

- Clarity and accessibility of presentation: some technical parts are dense; some figures are hard to read (Reviewers 3VjR, 88uW, m6da).
- Justification for excluding worst-case predictions in the smoothness definition (Reviewer PzFT).
- Applicability to other online problems, e.g., those exhibiting brittleness (Reviewer 88uW).
- Missing references to relevant works, e.g., Azar et al.'s results on flow-time scheduling with predictions and their use of multiplicative error (Reviewers PzFT, 88uW).


**Responses to reviews and updates in the revised version**

- **Improved clarity and readability.**
We have added intuition and explanatory text before technical content and reduced the complexities of Section 4. All details are in the appendix. We have revised figures to improve readability.
- **Justification for excluding worst-case predictions.**
We have justified our choice: compared to an alternative that includes worst-case predictions, our notion 1) is conceptually cleaner, 2) is more interpretable, and 3) yields stronger bounds.
- **Demonstrated broader applicability.**
We have demonstrated how our proposed smoothness definition applies to other online problems like one-way trading, a problem exhibiting brittleness, showing the broader impact of our work on other online problems beyond scheduling.
- **Added missing references.**
We have added and discussed Azar et al.'s work on flow-time scheduling with predictions and their multiplicative error metric. We have clarified connections and differences with our work.


**Broader impact**

Our work contributes to the story of learning-augmented algorithms by:

- Establishing that the informational advantages carried by predictions are bounded, showing what predictions can and cannot offer.
- Showing that, for some problems, asymptotically optimal consistency, robustness, and smoothness are simultaneously achievable, justifying the design of good learning-augmented algorithms.
- Demonstrating applicability beyond scheduling, thus offering a unified perspective on smoothness for other online problems.

We appreciate the reviewers' insights and positive assessment. We believe the revisions substantially improve the paper's clarity and impact. We hope that our responses address the concerns.

----

**To Area Chair**

We thank the Area Chair for overseeing the discussion. The reviewers consistently highlight the novelty of our refined definitions for the fundamental notions in learning-augmented algorithms, as well as the non-trivial, optimized, and in some cases tight smoothness bounds. After the revisions, we believe our work presents a clear and novel perspective for consistency, robustness, and smoothness. We show that smoothness lower bounds exist, i.e., predictions offer bounded informational advantages, but it is possible to have algorithms with (asymptotically) optimal consistency, robustness, and smoothness simultaneously. This result justifies good efforts in finding such algorithms.

Given a growing interest in learning-augmented algorithms in the machine learning community and the fundamental nature of our contributions, we believe the paper will be of strong interest to the ICLR audience. Your strong support would greatly help increase the visibility of our work (e.g., through an oral presentation). Giving an oral presentation at a top conference has long been a personal aspiration. We feel this work meets that standard.


Sincerely

Authors of Paper 5386

---

### Meta-Review · Area_Chair_DqqW · 2025-12-23

**Summary:**

The paper's primary contribution is the formalization of a smoothness metric that partitions the problem space based on the content of predictions. Using this metric, the authors derive improved or new lower bounds for three fundamental non-clairvoyant scheduling problems and provide matching or near-matching algorithms.

Strengths:
- Reviewers liked the conceptual novelty of distinguishing instances where predictions reveal information from those that do not.
- The shift to multiplicative error is natural and superior fit for scheduling problems

Weaknesses
- Multiple reviewers (3VjR, 88uW) mentioned that the technical definitions (specifically $R(I)$) and figures were difficult to parse.

Overall, the paper is a nice contribution and would be a nice addition to the ICLR program.

**Reviewer Concerns:**

All major concerns have been addressed.

**Reviewer Scores:**

I am counting the 2 as a 6 as reviewer PzFT explicitly mentioned that will raise the score to 6. The other reviewer likely would not change their score.

---

### Decision · Program_Chairs · 2026-01-26

Accept (Poster)